# DEEP GRAPH TRANSLATION

## ABSTRACT

The tremendous success of deep generative models on generating continuous data like image and audio has been achieved; however, few deep graph generative models have been proposed to generate discrete data such as graphs. The recently proposed approaches are typically unconditioned generative models which have no control over modes of the graphs being generated. Differently, in this paper, we are interested in a new problem named *Deep Graph Translation*: given an input graph, the goal is to infer a target graph by learning their underlying translation mapping. Graph translation could be highly desirable in many applications such as disaster management and rare event forecasting, where the rare and abnormal graph patterns (e.g., traffic congestions and terrorism events) will be inferred prior to their occurrence even without historical data on the abnormal patterns for this specific graph (e.g., a road network or human contact network). To this end, we propose a novel Graph-Translation-Generative Adversarial Networks (GT-GAN) which translates one mode of the input graphs to its target mode. GT-GAN consists of a graph translator where we propose new graph convolution and deconvolution layers to learn the global and local translation mapping. A new conditional graph discriminator has also been proposed to classify target graphs by conditioning on input graphs. Extensive experiments on multiple synthetic and real-world datasets demonstrate the effectiveness and scalability of the proposed GT-GAN.

## 1 INTRODUCTION

In recent years, deep learning on graphs has experienced a fast increase, especially for representation and discriminative tasks such as graph classification (Bruna et al., 2014; Defferrard et al., 2016; Kipf & Welling, 2017) and embedding (Grover & Leskovec, 2016; Hamilton et al., 2017). Beyond these successes in several applications, most recently researchers started to explore using deep generative models for graph synthesis on practical applications such as designing of new chemical molecular structure (Simonovsky & Komodakis, 2018) and social interaction modeling (You et al., 2018). This is because although the domain of generative graph models has a long history, traditional methods are highly dependent on the prior knowledge of the graph topological assumptions such as random graphs, scale-free graphs, and Kronecker graphs. But for many real-world problems the real topology are unknown, and hence we need to directly learn the generative models from the observations of the graphs and learn the multi-layer latent representations automatically.

This motivates the recent advances in deep generative models, some are domain dependent such as context-free grammars into the generative models (Kusner et al., 2017; Dai et al., 2018) while some others are generic (Simonovsky & Komodakis, 2018; Li et al., 2018; Samanta et al., 2018; Jin et al., 2018), most of which can only work on small graphs with 20 or fewer nodes. The existing approaches are typically unconditioned generative models, which typically only synthesize additional graphs directly following the distributions of the observation graphs and has no control over modes of the graphs being generated. However, in many important practical applications, it is usually crucial to guide the graph generation process by conditioning the model on additional information such as data from the different modalities.

In this paper, we are interested in generating a new graph conditioning on an input graph for the applications such as enterprise malicious-network detection, disaster management, and virus propagation synthesis. For example, trade secrets stored in the enterprise's computer network are vital to the company but is extremely difficult to defend if the attacks are from the internal employees' accounts (Neha, 2017). The theft-of-trade-secret behaviors involve highly sophisticated activities in

terms of malicious authentication paths over cyber-networks. And the recognition of them requires to learn from rich historical attack examples, which barely can be available since these accounts always being monitored well.

Although theft behaviors on cyber-networks have the similar purpose, the behaviors from different accounts look quite different because of various sizes and structures of computer networks accessible by distinct accounts: some are administrators each with 100+ computers while some only can access less than five. This requires us to learn the generic distribution of theft behaviors from historical attacks and synthesize the possible malicious authentication graphs for the other accounts conditioning on their current computer networks.

We formulate this type of new problems as *Deep Graph Translation*, which aims at translating a graph with one modality to a new one with other modality using deep neural networks architecture. This problem is analogical to image-to-image translation in image processing (Isola et al., 2017) and language translation in natural language processing (Yang et al., 2017). Unfortunately, similar to the consensus that the existing image or text generation methods are not appropriate to be directly applied to graph generation (Simonovsky & Komodakis, 2018), existing image or text translation methods cannot be directly applied to Deep Graph Translation problems. It is worth noting that, although training an encoder-decoder model on graphs under supervised setting could generate a new graph given a graph input, this will only produce a single deterministic graph based on a graph input. However, as a graph synthesis task, we are instead interested in learning a distribution of graphs conditioning on a graph input, which naturally are non-deterministic outputs capturing underlying properties of the graphs such as scale-free and other implicit properties. To achieve this, there are critical challenges in 1) learning to distill shared generic graph translation patterns over graphs with different structures, and 2) assimilating the learned graph translation mapping to generate possible target graphs conditioning on an input graph with a new structure.

To address the aforementioned challenges, in this paper, we present a novel framework for deep graph translation with graph-translation generative adversarial nets (GT-GAN). Different from existing ordinary GANs (Goodfellow et al., 2014) that learn a generative model of data distribution, GT-GANs learn a conditional graph generative model, which is a graph translator that is conditioned on input graph and generates the associated target graph. The GT-GAN consists of a graph translator and a novel conditional graph discriminator. The graph translator includes three parts - graph convolutional layers, novel graph deconvolutional layers, and a graph U-net, which collectively address the second challenge by jointly leveraging local and global information on graphs. Moreover, because the fidelity of the generated graphs needs to be discriminated depending on the specific input graph, we propose a new conditional graph discriminator that is built on GCNN but conditioning on the input graphs. Our contributions are summarized as follows:

- We formulate for the first time a graph translation problem. We further develop the first deep framework for graph translation by proposing a new conditional graph GAN.

- We propose the first graph deconvolutional layers for the graph translation, in order to generate target graph while preserving the local information from input graph.

- Our proposed GT-GAN is scalable with at most quadratic computational complexity and memory consumption in terms of the number of nodes in a graph, making it suitable for at least modest-scale graph translation problems.

- Our GT-GAN is highly extensible where underlying building blocks, GCNN (Graph Convolution Nueral Network) and distance measure in discriminator, can be replaced by other techniques such as (Kipf & Welling, 2017; Arjovsky et al., 2017) or their extensions.

- Extensive experiments on both synthetic and real-world application data demonstrate that GT-GAN is capable of generating graphs close to ground-truth target graphs and significantly outperforms other VAE-based models in terms of both effectiveness and efficiency.

## 2 RELATED WORKS

Deep learning techniques dealing with graphs is a new trending topic in recent years.

**Neural networks on graph representation learning**. In recent years, a surge of research in neural networks on graphs are generally divided into two categories: Graph Recurrent Networks (Gori

et al., 2005; Scarselli et al., 2009; Li et al., 2016) and Graph Convolutional Networks (Niepert et al., 2016; Mousavi et al., 2017; Defferrard et al., 2016; Kawahara et al., 2017; Nikolentzos et al., 2017; Cao et al., 2016; Kipf & Welling, 2017; Xu et al., 2018). Graph Recurrent Networks originates from early works for graph neural networks proposed by Gori et al. (Gori et al., 2005) and Scarselli et al. (Scarselli et al., 2009) based on recursive neural networks. Graph neural networks were then extended using modern deep leaning techniques such as gated recurrent units (Li et al., 2016). Another line of research is to generalize convolutional neural networks from grids (e.g., images) to generic graphs. Bruna et al. (Bruna et al., 2014) first introduces the spectral graph convolutional neural networks, and then extended by Defferrard et al. (Defferrard et al., 2016) using fast localized convolutions, which is further approximated for an efficient architecture for a semi-supervised setting (Kipf & Welling, 2017). However, most of them are mainly developed for a single undirected large graph. To consider the various topology of different graphs, our graph encoder draws inspiration from (Kawahara et al., 2017), we present a graph CNN consisting of a sequence of new layers for the convolution operations on edge-to-edge, edge-to-node, and node-to-graph for directed graphs.

**Graph generation**. Until now the deep models proposed for graph generation can be generally categorized into two classes: 1) domain-specific models and 2) generic models. Domain-specific models (Gómez-Bombarelli et al., 2016; Kusner et al., 2017; Dai et al., 2018; Jin et al., 2018) typically take sequence inputs (such as SMILES for representing molecules) and generate other sequences or parse trees from context-free grammars or sub-graphs by utilizing the collection of valid components. However, these methods are not general for being widely applied.

Generic graph generation can handle general graphs that are not restricted to specific applications, which is more relevant to this paper. Existing works are almost proposed in the most recent year, which are based on VAE (Simonovsky & Komodakis, 2018; Samanta et al., 2018), generative adversarial nets (GAN) (Bojchevski et al., 2018) and others (Li et al., 2018; You et al., 2018). Specifically, Li et al. (2018) proposed a graph net that generates nodes and edges sequentially to form a whole graph, which is sensitive to the generation order and time-consuming for the large graphs. Simonovsky & Komodakis (2018) and Samanta et al. (2018) both propose new variational autoencoders in parallel for whole graph generation, though they typically only handle very small graphs (e.g., with $\leq 50$ nodes) and cannot scale well in both memory and runtime for large graphs. Different from the above methods, GraphRNN represents graphs as sequences using different node ordering, and then builds an autoregressive generative model on these sequences with LSTM model. Compared to existing methods, GraphRNN shows its good scalability in terms of graph size, which is also comparable with our GT-GAN but with much higher runtime due to its large constants in the runtime complexity and memory consumption as we will show in the experimental section.

**Conditional generative adversarial nets**. Generative adversarial networks (GANs) (Goodfellow et al., 2014) significantly advanced the state of the art over the classic prescribed approaches like mixtures of Gaussians for image and 3D object generation (Karras et al., 2017). To further empower GANs with the capability to generate objects under certain conditions, recently conditional GANs have been proposed to condition GANs on discrete labels (Mirza & Osindero, 2014), text (Reed et al., 2016), and especially images (Wang & Gupta, 2016). Conditional GANs enable multi-modal data fusion to generate images according to specific annotations, text labels or translate an input image to an output image. For example, many important applications in image processing domain using conditional GANs include image-to-image translation (Isola et al., 2017), inpainting (Pathak et al., 2016), and style transfer (Zhu et al., 2017). However, to the best of our knowledge, there is no work focusing on graph-to-graph translation problem.

## 3 GT-GAN

The problem formulation of graph translation is first presented. Then we present the method GT-GAN and describe its graph translator and conditional graph discriminator in details.

### 3.1 DEEP GRAPH TRANSLATION BY GT-GAN

This paper focuses on the graph translation from an *input graph* to a *target graph*. We define an *input graph* $X$: $G_X = (V, E, W)$ as a directed weighted graph such that $V$ is the set of $N$ nodes, $E \subseteq V \times V$ is the set of directed edges, and $W \in \mathbb{R}^{N \times N}$ is a *weighted adjacency matrix* holding the set of weights for the corresponding edges. Denote $e_{i,j} \in E$ as an edge from the node $v_i \in V$ to

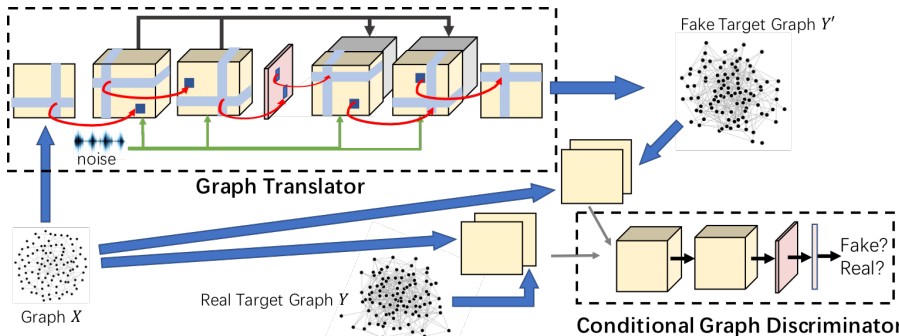

Figure 1: Architecture of GT-GAN consisting of graph translator and conditional graph discriminator. New graph encoder and decoder are specially designed for graph translation.

$v_j \in V$ and thus $W_{i,j} \in W$ denotes the corresponding weight of the edge $e_{i,j}$. Similarly, we define a *target graph* $G_Y = (V', E', W')$ as a directed weighted graph. Typically we focus on learning the translation from one topological patterns to the other with the same set of nodes and hence we have $V = V'$. Additionally, the weights of edges are nonnegative such that $W_{i,j} \geq 0$ and $W'_{i,j} \geq 0$. We define a new problem named **graph translation**, where we focus on learning a translator from an input graph $G_X \in \mathcal{G}_X$ and a random noise $U$ to a target graph $G_Y \in \mathcal{G}_Y$. The noise $U$ is introduced by the dropout function (Seltzer et al., 2013) in each convolution and deconvolution layer as shown in green lines of Fig. 1 . The translation mapping is denoted as $T : U, G_X \to G_Y$, where $\mathcal{G}_X$ and $\mathcal{G}_Y$ denote the domains of input and target graphs.

To address this issue, we proposed the Graph-Translation GAN (GT-GAN) that consists of graph translator $T$ and conditional graph discriminator $D$, as shown in Fig. 1. $T$ is trained to produce target graphs that cannot be distinguished from "real" graphs by $D$, that is, distinguishes the produced target graph $G_{Y'} = T(G_X, U)$ from the real one $G_Y$ based on the current input graph $G_X$. $T$ and $D$ undergo an adversarial training process based on input and target graphs shown as below:

$$\mathcal{L}(T, D) = \mathbb{E}_{G_X, G_Y}[\log D(G_Y|G_X)] + \mathbb{E}_{G_X, U}[\log(1 - D(T(G_X, U)|G_X))] \quad (1)$$

where $T$ tries to minimize this objective against an adversarial $D$ that tries to maximize it, i.e. $T^* = \arg\min_T \max_D \mathcal{L}(T, D)$. Previous approaches in image domain have found it beneficial to mix the GAN loss with L1 loss (Isola et al., 2017). Thus, an L1 loss is applied to the weight adjacent matrix of generated target graphs and real target graphs:

$$\mathcal{L}_{l1}(T) = \mathbb{E}_{G_X, G_Y, U}[\|G_Y - T(G_X, U)\|_1] \quad (2)$$

The training process is a trade-off game between $\mathcal{L}_{l1}$ and $\mathcal{L}(T, D)$, which jointly enforces $T(G_X, U)$ and $G_Y$ to follow a similar topological pattern but may not necessarily the same. Specifically, $\mathcal{L}_{l1}$ makes $T(G_X, U)$ share the same rough outline of sparsity pattern with $G_Y$, while under this outline, $\mathcal{L}(T, D)$ allows the $T(G_X, U)$ to vary to some degree. Thus, the final objective is:

$$T^* = \arg\min_T \max_D \mathcal{L}(T, D) + \mathcal{L}_{l1}(T) \quad (3)$$

## 3.2 GRAPH TRANSLATOR

We propose a new graph translator by extending the state-of-the-art graph convolution and proposing new graph deconvolution layers.

Different and more difficult than graph generation designed only for learning the distribution of graph representations, graph translation learns not only the latent graph presentation but also the generic translation mapping from input graph to the target graph simultaneously. However, graph representations vary according to different graph samples while the translation mapping between input-target pairs remains consistent across different pairs of samples. Moreover, different from graph generation which focuses on the graph level distribution, graph translation focus on learning the translation mapping both in graph level (e.g., general transformation in graph metrics) and in local level (e.g., specific nodes have specific transformations according to its local properties).

To address the first challenge, we propose to leverage skip-net structure (Ronneberger et al., 2015) so that the sample-specific representations can be directly passed over through skip connection to decoder's layers while the sample invariant mapping will be learned in the encoder-decoder structure.

More importantly, to address the second challenge, we extend the state-of-the-art graph convolution and propose new graph deconvolution that preserves both global and local information during graph translation. The architecture of graph translator is illustrated in Fig. 1, where the input graph first undergoes two "directed edge-to-edge convolution' operations to encode the higher-order topological information and then is embedded into node representation by a "directed edge-to-node convolution" operation. Then, a novel graph decoder based on graph deconvolution has been proposed, including one "directed node-to-edge deconvolution" operation and two "directed edge-to-edge deconvolution" operations. The details are introduced in the following.

### 3.2.1 DIRECTED GRAPH CONVOLUTIONS

Some recent works focus on generalizing image convolution into graph convolution, but as pointed out in (Kawahara et al., 2017), most of them try to embed the graphs based on the signals assigned on the nodes with the fixed graph topology. The work that can handle encoding the edge structure typically focus on undirected graphs but are not immediately applicable to directed graphs.

Therefore, to solve this issue, we propose a graph encoder that can admit directed graph convolution based on (Kawahara et al., 2017). In a directed weighted graph, each node could have in-edge(s) and out-edge(s), which can be convoluted respectively for each node. Specifically, denote the $A^{l,m} \in \mathbb{R}^{N \times N}$ as the weighted adjacency matrix for $l$th layer in $m$th feature map and $A_{i,j}^{l,m}$ is for the edge $e_{i,j}$. Let $A^{1,1} \equiv W$ denote the weighted adjacency matrix of the input graph. $\Phi_{l,m} \in \mathbb{R}^{1 \times N}$ and $\Psi_{l,m} \in \mathbb{R}^{N \times 1}$ are the incoming and outgoing kernels of $l$th layer of $m$th feature map for a node, respectively. We define a graph convolution over its in-edge(s) as the weighted sum over all the weights of its incoming edges: $f_{l,m,n,j}^{(\text{in})} = \Phi_{l,m,n} \cdot A_{\cdot,j}^{l,m}$, where $n$ denotes the $n$-th feature map in $(l + 1)$-th layer. Similarly, we define the graph convolution over the out-edge(s) as $f_{l,m,n,i}^{(\text{out})} = A_{i,\cdot}^{l,m} \cdot \Psi_{l,m,n}$. And thus the *directed edge-to-edge convolution* is defined as follows:

$$A_{i,j}^{l+1,n} = \sigma\left(\sum\nolimits_{m=1}^{M_l} (f_{l,m,n,i}^{(\text{in})} + f_{l,m,n,j}^{(\text{out})})\right) \quad (4)$$

where $A_{i,j}^{l+1,n}$ refers to the value in position $i,j$ of edge $n$th feature map in the $(l + 1)$th layer. $M_l$ refers to the number of feature maps in the $l$th layer. The two components refers to out-direction and in-direction features as talked above. $\sigma(\cdot)$ refers to activation function that can be set as linear, or ReLu (Veličković et al., 2017) when the edge weights are assumed nonnegative.

The *directed edge-to-node convolution* embeds each edge feature map into a vector which encodes all the incoming and outgoing edges of a node into a value from various combinations:

$$A_i^{l+1,n} = \sigma\left(\sum\nolimits_{m=1}^{M_l} (f_{l,m,n,i}^{(\text{in})} + f_{l,m,n,i}^{(\text{out})})\right) \quad (5)$$

where $A_i^{l+1,n} \in \mathbb{R}^{N \times 1}$ denotes the $i$th node representation (i.e., the 4th layer in graph translator in Fig. 1) under $n$th feature map.

### 3.2.2 GRAPH DECONVOLUTIONS

The graph convolution encodes the input graph into node representation with highly condensed knowledge on the higher-order neighborhood pattern of nodes. Next, decoder part aims to decode the node representation back to target graph as shown in Fig. 2. Graph deconvolution decodes the single node (or edge) information to its incoming and outgoing neighbor nodes and edges, which is a reverse process of graph convolution. Currently, to the best of our knowledge, there is no work on directed graph deconvolution. The techniques that can generalize the image deconvolution to graph domain is highly imperative and nontrivial. As shown in Fig. 3, this paper proposes novel graph deconvolution techniques including "node-to-edge deconvolution" layers and "edge-to-edge deconvolution" layers, which are described as follows:

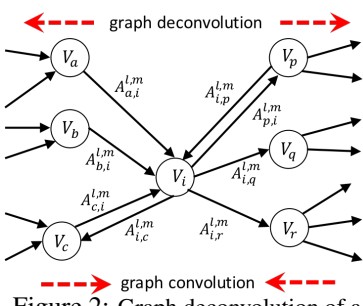

Figure 2: Graph deconvolution of a node

**node-to-edge deconvolution**. Different from images, "directed node-to-edge deconvolution" requires to decode the "local neighborhood" pattern from a node through graph topology including both incoming and outgoing connections. This calls for a reverse process of "directed edge-to-node convolution". As shown in Fig. 3(b), each node is decoded

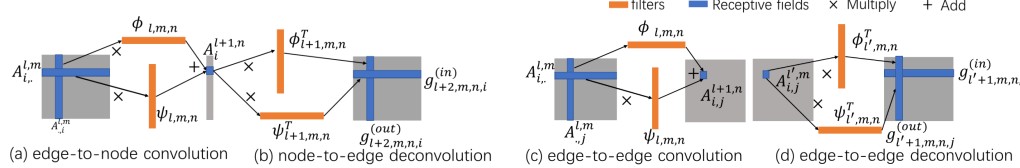

Figure 3: Matrix operations for graph convolution and graph deconvolution. In convolution operations, we need to utilize row filter to convolute "incoming" edges and column filter for "outgoing" edges. However, in deconvolution operations, we have to utilize the transposed filters, namely column filter to decode for "incoming" edges and row filter to decode for "outgoing" edges.

back to its "incoming" and "outgoing" edges by two transposed filters. Specifically, each node $A_i^{l,m}$ in the $m$th feature map in $l$th layer is multiplied by these two filters $\Phi_{l,m,n}^T$ and $\Psi_{l,m,n}^T$ respectively to get the decoded information in terms of $g_{l+1,m,n,i}^{(in)} = \Phi_{l,m,n}^T \cdot A_i^{l,m}$ and $g_{l+1,m,n,i}^{(out)} = A_i^{l,m} \cdot \Psi_{l,m,n}^T$ for the $n$th feature maps in $(l+1)$th layer, as shown in Fig. 3(b). Here $\Phi_{l,m,n}^T \in \mathbb{R}^{1 \times N}$ is a row filter for decoding "outgoing" edges while $\Psi_{l,m,n}^T \in \mathbb{R}^{N \times 1}$ is a column filter for decoding "incoming" edges. Hence they are respectively vertical to the corresponding filters for convolution, as shown in Fig. 3. Based on such deconvolution process, all nodes in $l$th layer will be decoded into corresponding pairs of a column and a row, which will be aggregated into the feature map for the $(l + 1)$th layer. Mathematically, each element of $A_{i,j}^{l+1,n}$ in the $n$th feature map is calculated as:

$$A_{i,j}^{l+1,n} = \sum_{m=1}^{M} \sigma([\underbrace{\Phi_{l,m,n}^T \cdot A_j^{l,m}}_{g_{l+1,m,n,j}^{(in)}}]_i + [\underbrace{A_i^{l,m} \cdot \Psi_{l,m,n}^T}_{g_{l+1,m,n,i}^{(out)}}]_j) \tag{6}$$

where $[\cdot]_i$ means the $i$th element of a vector.

**edge-to-edge deconvolution**. The decoded feature map from "node-to-edge deconvolution" still encompasses higher-order connectivity knowledge, which will be further released and translated back to the neighborhood of "incoming" and "outgoing" incident edges by "edge-to-edge deconvolution". We propose "edge-to-edge deconvolution" which will decode each edge back to its "incoming" edges (i.e., the incoming edges of its source node) and "outgoing" edges (i.e., the outgoing edges of its target node) incident to it by two transposed filters. As shown in Fig. 3(d), each edge representation $A_{i,j}^{l',m}$ in $m$th feature map in $l'$th layer will be multiplied by both the filters $\Phi_{l',m,n}^T$ and $\Psi_{l',m,n}^T$ respectively to get its relevant decoded information in terms of $g_{l'+1,m,n,i}^{(in)} = \sum_{m=1}^{M} \Phi_{l',m,n}^T \cdot A_{m,i}^{l',m}$ and $g_{l'+1,m,n,i}^{(out)} = \sum_{m=1}^{M} A_{i,m}^{l',m} \cdot \Psi_{l',m,n}^T$ for $n$th feature maps in $l' + 1$ layer. Here $\Phi_{l',m,n}^T$ is a row filter for decoding "outgoing" edges while $\Psi_{l',m,n}^T$ is a column filter for decoding "incoming" edges, as shown in Fig. 3. Hence, based on such deconvolution process, all elements (i.e., edges) in $l'$th layer will be decoded into corresponding pairs of a column and a row, which will be finally aggregated into the feature map for the $(l' + 1)$th layer. Mathematically, each element of $A_{i,j}^{l'+1,n}$ in the $n$th feature map is calculated as:

$$A_{i,j}^{l'+1,n} = \sigma([\underbrace{\sum_{m=1}^{M} \Phi_{l',m,n}^T \cdot A_{m,i}^{l',m}}_{g_{l'+1,m,n,i}^{(in)}}]_j + [\underbrace{\sum_{m=1}^{M} A_{j,m}^{l',m} \cdot \Psi_{l',m,n}^T}_{g_{l'+1,m,n,j}^{(out)}}]_i) \tag{7}$$

In sum, graph deconvolution can be seen as a transposed operation of graph convolution, as shown in Fig. 3, which is analogical to the relationship between image convolution and deconvolution[1].

### 3.3 Conditional Graph Discriminator

When identifying the "translated" target graph from the "real" ones, the input graph will also need to be considered. This new problem requires the discriminator to take two graphs as inputs, namely a target graph and an input graph, and classify the joint patterns of them into a class label. Instead of identifying whether the generated graphs look similar to the real graphs, this problem requires to identify whether the translations look similar, which encodes a second-order change between input and target graphs. The challenge is not only learning the topological representation of each graph but also need to learn the correspondence between the two graphs and their nodes globally and locally. To address this, we propose *conditional graph discriminator* (CGD) which leverages

---

[1]This is why "deconvolution" is well-recognized to be better named as "transposed convolution"

"paired" directed edge-to-edge layers as shown in Fig. 1. Specifically, the input and target graphs are both ingested by CGD and stacked into a $N \times N \times 2$ tensor which can be considered as a 2-channel weighted adjacency matrix of a multi-graph. Then each of the channels is mapped to its corresponding feature maps and then the separated directed edge-to-edge layers. Then the edge-to-node layer is again applied to obtain node representations, which is then mapped to graph embedding by a fully-connected layer. Finally, a softmax layer is implemented to distinguish the real graph and fake graph. When the model is well trained, the discriminator cannot distinguish the generated target graphs and real target graphs, the output probabilities of all graphs should be around 0.5.

## 3.4 ANALYSIS ON TIME AND MEMORY COMPLEXITY

The graph encoder and decoder shares the same time complexity. Without loss of generalization, we assume all the layers (except input and output) have the same number of feature maps as $M_0$. $P$ is the length of the fully connected layer. Then, the worst-case total complexity of GT-GAN (i.e., when the weighted adjacency matrix is dense) is $O(9N^2M_0^2 + 3N^2M_0^2 + N^2M_0P)$, where the first, second, and third terms are for "directed edge-to-edge convolutions", "directed edge-to-node convolutions", and fully connected layers in conditional graph discriminator, respectively. Similarly, the total memory consumption of GT-GAN is $O((9NM_0^2 + 9N^2M_0) + (3NM_0^2 + 3NM_0) + (N^2M_0P + P))$. Compared to the existing works (Simonovsky & Komodakis, 2018; Samanta et al., 2018) that can only scale to small size of graphs (up to $|V| = 20$), our GT-GAN is able to provide a scalable (i.e., $O(N^2)$) algorithm that can generate general graphs.

## 4 EXPERIMENT

In this section, extensive experiments on graph translation for the proposed methods on two synthetic and one real-world dataset are conducted and analyzed. The proposed GT-GAN demonstrated outstanding performance in both effectiveness and scalability by multiple metrics. All experiments are conducted on a 64-bit machine with Nvidia GPU (GTX 1070,1683 MHz, 8 GB GDDR5). Our code and data are available at `https://github.com/anonymous1025/Deep-Graph-Translation-`.

### 4.1 EXPERIMENTAL SETTINGS

#### 4.1.1 DATASETS

**Synthetic Datasets:** Two groups of synthetic datasets are used to validate the performance of the proposed GT-GAN: Scale-free graphs set and Poisson-random graphs set. Each group has five datasets with different graph sizes (i.e., number of nodes): 10, 20, 50, 100, and 150. Each dataset consists of 5,000 pairs of input and target graphs: 2500 pairs of graphs are used for training the graph translator and the remaining 2500 pairs are used to validate the performance of graph translation on input graphs. The generation rule for each group of datasets is as follows.

*Synthetic Datasets 1: Scale-free Graph Datasets.* Each input graph is generated as a directed scale-free network, which is a network whose degree distribution follows power-law property (Bollobás et al., 2003). A node will by selected as target node with probability proportional to its in-degree, which will be linked to a new source node with probability of 0.41. Similarly, a node will by selected as source node with probability proportional to its out-degree, which will be linked to a new target node with probability of 0.54. Then, a corresponding target graph is generated by adding $m$ ($m$ equals the number of nodes of the input graph) edges between two nodes. Thus, both input and target graphs are scale-free graphs.

*Synthetic Datasets 2: Poisson-random Graph Datasets.* Each input graph is generated by (Krapivsky & Redner, 2001) which is a directed growing random network. Then for an input graph with $|E|$ edges, we randomly add $k|E|$ edges on it to form the target graph, where $k$ follows the Poisson distribution with the mean of 5.

**Real World Datasets: User authentication Graph Datasets**. This dataset includes the authentication activities of 97 users on their accessible computers and servers in an enterprise computer network (Kent, 2015). Over time, each user account logs on one computer from another, which forms a directed weighted graph called authentication graph: nodes represent computers while the directed edges and weights represent the authentication activities and the frequencies. The goal is to forecast and synthesize the future potential malicious authentication graphs of the users without any historical malicious behaviors, by the graph translator from normal to malicious graph trained based

on the users with historical malicious-behavior records. Here, each authentication graph records the user behavior in thirty minutes. In total, we have 315 samples pairs, and each of them consists of a pair of authentication graphs with and without malicious (e.g., hacking) behavior. We perform 3-fold cross-validation for training and testing.

### 4.1.2 COMPARISON METHODS

Since there is no existing work on deep graph translation, we compare against current state-of-the-art baselines that can generate general graphs, including 1) GraphRNN (You et al., 2018): a newly graph generation method based on sequential generation with LSTM model; 2) GraphVAE (Simonovsky & Komodakis, 2018): a probability-based graph generation method for small graphs; 3) GraphGMG (Li et al., 2018): a framework based upon graph neural networks for small single graphs; 4) RandomVAE (Samanta et al., 2018); and 5) S-Generator: supervised deterministic graph generation using only our graph translator with L1 loss. All the comparison methods are directly trained by the malicious graphs without the conditions of input graphs as they can only do graph generation instead of translation. When applying these methods, which are for undirected graphs, to directed graphs problem, the adjacent matrix of directed graphs is transformed from size $N$ to size $2N$. Due to the limited scalability of the above methods, GraphRNN is tested with graph size within 150. GraphGMG, GraphVAE is tested within size 10 and RandomVAE is tested on graphs within size 50. All the comparison methods are trained through ADAM optimization algorithm. The learning rate of GraphGMG, GraphVAE, and RandomVAE is 0.001 and the learning rate of GraphRNN is 0.003.

## 4.2 PERFORMANCE

The scalability of different methods on different datasets and model effectiveness evaluation by direct evaluations and indirect evaluations are given.

### 4.2.1 MODEL SCALABILITY ANALYSIS

Unlike most deep graph generative models which only focus on very small graphs (e.g., $< 40$ nodes), the proposed GT-GAN can scale to graphs of larger size. To demonstrate it, Fig. 4 illustrates the scalability of GT-GAN and two baseline methods (GraphRNN and RandomVAE) in terms of memory consumption and computational time, respectively. Other comparison methods are not tested since they can only deal with small graphs with node number less than 20. As shown in Fig. 4, when the graph size increases up to 50, the memory consumption of the GT-GAN maintains almost constant and computational time grows slowly, both of which are less than 1500MB and 300 seconds, respectively. In contrast, the memory consumption and computational time of RandomVAE increases super-linearly as the graph size increases, making it hard to scale even to the graph size of 30. Interestingly, memory consumption and runtime of GraphRNN increases slightly as the graph size increases, but their costs are much higher than GT-GAN with almost two times in memory requirement and ten times in runtime.

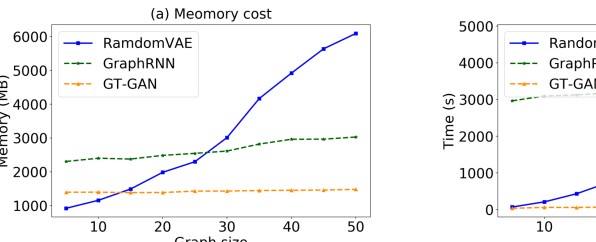

Figure 4: Scalability plots on memory and time cost of GT-GAN, RandomVAE and GraphRNN

### 4.2.2 DIRECT EVALUATION OF MODEL EFFECTIVENESS

To directly verify if GT-GAN indeed discover the underlying ground-truth translation rules between input and target graphs, the sparsity similarity between generated target graphs and real target graphs are measured on various metrics.

**Results on Scale-Free Graphs**: For scale-free graph datasets, the metrics are: Jensen-Shannon distances (JS) (Lin, 1991), Hellinger Distance (HD) (Beran, 1977), Bhattacharyya Distance (BD) (Basseville, 1989) and Wasserstein Distances (WD) (Rüschendorf, 1985), which are most commonly used to measure the distance between two distributions. Table 1 shows the distances between generated target graphs and real target graphs in terms of node degree on the above metrics. The "Inf" in Tabel 1 represents the distance more than 1000. The proposed GT-GAN outperforms others

in all metrics with smaller distances, especially in Bhattacharyya distance (by average 15%) and Hellinger distance for all graph sizes. S-Generator is the second best methods in terms of distance evaluation since it is a supervised learning method. Fig. 5 directly shows the node degree distribution curve of some generated and real target graphs by GT-GAN. The curves of the generated graphs follow power-law rule correctly and become closer to the real target graphs when graph size increases, which is consistent with Table 1. This again demonstrates that our end-to-end graph translator indeed can accurately discover the underlying translation by scale-free model. Due to space limits, many other examples can be found in Appendix A in supplementary materials. Similar observations in the direct evaluation results (e.g. average degree, repository and density) of Poisson random graphs and user authentication graphs can be found in Appendix B and C.

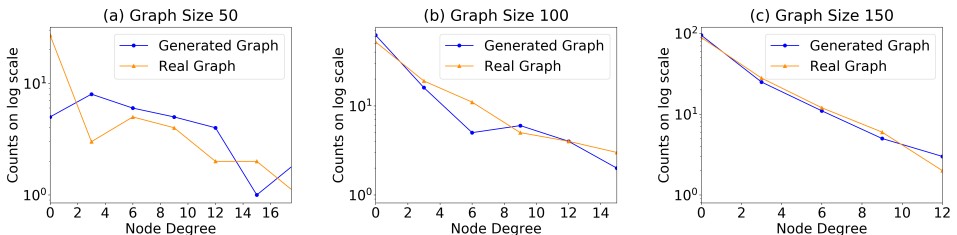

Figure 5: Examples of node degree distributions of generated and target graphs for scale free graphs

Table 1: Node degree distribution distance between the generated and real graphs for scale free graphs

| Graph size | Methods | Jensen-Shannon | Hellinger | Bhattacharyya | Wasserstein |
|---|---|---|---|---|---|
| 10 | Random-VAE | 0.42 | **0.98** | Inf | 7.58 |
| | GraphRNN | 0.47 | **0.98** | Inf | 1.64 |
| | GraphVAE | 0.67 | 1.00 | Inf | 2.85 |
| | GraphGMG | 0.43 | **0.98** | Inf | 1.69 |
| | S-Generator | **0.35** | 0.98 | 3.45 | 0.80 |
| | GT-GAN | **0.35** | 0.98 | **3.44** | **0.77** |
| 20 | RandomVAE | 0.51 | 0.97 | Inf | 1.74 |
| | GraphRNN | 0.50 | 0.98 | Inf | 1.44 |
| | S-Generator | 0.36 | **0.96** | 2.84 | 0.67 |
| | GT-GAN | **0.35** | **0.96** | **2.74** | **0.66** |
| 50 | Random-VAE | 0.69 | 0.99 | Inf | 7.99 |
| | GraphRNN | 0.49 | 0.94 | Inf | 1.44 |
| | S-Generator | **0.31** | 0.90 | 1.70 | **0.34** |
| | GT-GAN | 0.43 | **0.89** | **1.66** | 2.43 |
| 100 | GraphRNN | 0.48 | 0.88 | Inf | 0.90 |
| | S-Generator | **0.14** | 0.68 | 0.64 | **0.30** |
| | GT-GAN | 0.15 | **0.43** | **0.24** | 0.31 |
| 150 | GraphRNN | 0.42 | 0.74 | Inf | 0.95 |
| | S-Generator | 0.08 | 0.31 | **0.11** | 0.29 |
| | GT-GAN | **0.07** | **0.30** | **0.11** | **0.27** |

### 4.2.3 INDIRECT EVALUATION OF MODEL EFFECTIVENESS

Graph classification usually faces label imbalance issue. Here, beyond label imbalance, we are interested in "label missing" which is more challenging. For example, we want to classify whether an authentication graph of a user is malicious (positive) or normal (negative), but this user does not have any malicious records before. In this extremely difficult task, the target graphs (i.e., malicious graphs) generated by graph translator will be utilized as positive samples to train the classifier.

More generally, the validation set introduced in Section 4.1.1 of each dataset will be further split evenly into two subsets. The first subset is for training a graph classifier based on model proposed byNikolentzos et al. (2017), using only the negative samples (i.e., the input graphs) plus the target graphs generated by each comparison method. In addition, we have also trained a "gold standard" classifier based on input graphs and *real* target graphs. Finally, the second subset contains both the input graphs and real target graphs for validating the classifiers trained above. Therefore, gold standard method acts as the best-possible-performer to evaluate all the different generative models on how real the graphs they generate. The evaluation process is shown in Fig. 10 in Appendix D.

**Results on Scale Free Graphs**: Table 2 shows the average results of graph classifiers: Precision, Recall, AUC, and F1-measure on recognizing real target graphs by being trained with generated target graphs on different methods. For small graphs (e.g., nodes less then 20), the power-law property of scale-free networks is less obvious compared to the larger size graphs, which may explain why the tasks on smaller scale-free graphs are more difficult. As shown in Table2, the performance corresponding to GT-GAN is very close to the "Gold Standard" with average difference of 6%, 12%,

13% and 6% on precision, recall, AUC and F1 respectively, and outperforms other methods by 32%, 7.6%, 12.6%, and 30.8% on four metrics respectively.

**Results on Poisson Random Graphs**: Results for indirect evaluation on Poisson Random graphs are listed in Table 3(P=Precision, R=Recall, F1=F1-measure). Though the Poisson random task is easier than scale-free graphs, the GT-GAN still outperforms others significantly, especially on AUC results by around 13.4% on average, and its performance is highly close to the "Gold Standard" with average difference of 2.6%, 3.2%, 1.2% and 3.2% on precision, recall, AUC and F1 respectively.

Table 2: Indirect evaluation for scale-free graphs

| Size | Method | P | R | AUC | F1 |
|---|---|---|---|---|---|
| | RandomVAE | 0.83 | 0.29 | 0.31 | 0.42 |
| | GraphRNN | 0.31 | 0.11 | 0.49 | 0.16 |
| | GraphVAE | 0.75 | 0.23 | **0.65** | 0.35 |
| 10 | GraphGMG | 0.42 | 0.12 | 0.49 | 0.18 |
| | S-Generator | 0.46 | **0.83** | 0.43 | 0.59 |
| | GT-GAN | **1.00** | 0.50 | 0.52 | **0.67** |
| | Gold Standard | 0.81 | 0.74 | 0.82 | 0.77 |
| | RandomVAE | 0.50 | 1.00 | **0.54** | 0.66 |
| | GraphRNN | 0.67 | 0.12 | 0.50 | 0.21 |
| 20 | S-Generator | 0.50 | 1.00 | 0.50 | 0.67 |
| | GT-GAN | **1.00** | 0.50 | 0.50 | **0.67** |
| | Gold Standard | 0.76 | 0.67 | 0.72 | 0.71 |
| | RandomVAE | 0.89 | 0.67 | 0.84 | 0.76 |
| | GraphRNN | 0.52 | 0.53 | 0.70 | 0.52 |
| 50 | S-Generator | 0.50 | 1.00 | 0.37 | 0.67 |
| | GT-GAN | **0.93** | 0.82 | **0.94** | **0.87** |
| | Gold Standard | 0.94 | 0.90 | 0.97 | 0.91 |
| | GraphRNN | 0.61 | 0.65 | 0.67 | 0.60 |
| 100 | S-Generator | 0.50 | **1.00** | 0.50 | 0.67 |
| | GT-GAN | **0.72** | 0.69 | **0.68** | **0.70** |
| | Gold Standard | 0.99 | 0.61 | 0.81 | 0.75 |
| | GraphRNN | 0.73 | **0.92** | 0.92 | 0.81 |
| 150 | S-Generator | **1.00** | 0.50 | 0.50 | 0.67 |
| | GT-GAN | 0.94 | 0.79 | **0.96** | **0.86** |
| | Gold Standard | 0.99 | 0.93 | 0.96 | 0.95 |

Table 3: Poisson random graphs indirect evaluation

| Size | Method | P | R | AUC | F1 |
|---|---|---|---|---|---|
| | RandomVAE | 0.98 | 0.75 | **0.99** | 0.85 |
| | GraphRNN | 0.98 | 0.99 | **0.99** | **0.98** |
| | GraphVAE | 0.98 | 0.92 | 0.97 | 0.94 |
| 10 | GraphGMG | 0.98 | 0.98 | 0.98 | **0.98** |
| | S-Generator | 0.50 | **1.00** | 0.50 | 0.66 |
| | GT-GAN | **1.00** | 0.87 | 0.94 | 0.90 |
| | Gold Standard | 0.99 | 1.00 | 1.00 | 0.99 |
| | RandomVAE | **1.00** | 0.70 | 0.99 | 0.82 |
| | GraphRNN | **1.00** | **1.00** | **1.00** | **1.00** |
| 20 | S-Generator | **1.00** | **1.00** | **1.00** | **1.00** |
| | GT-GAN | **1.00** | 0.99 | **1.00** | 0.99 |
| | Gold Standard | 0.99 | 1.00 | 1.00 | 0.99 |
| | RandomVAE | 0.93 | 0.46 | **1.00** | 0.63 |
| | GraphRNN | **1.00** | 0.99 | 0.99 | 0.99 |
| 50 | S-Generator | 0.49 | 0.98 | 0.35 | 0.65 |
| | GT-GAN | **1.00** | 0.99 | **1.00** | **0.99** |
| | Gold Standard | 0.99 | 1.00 | 1.00 | 0.99 |
| | GraphRNN | **1.00** | 0.99 | **1.00** | **0.99** |
| 100 | S-Generator | 0.50 | **1.00** | 0.51 | 0.66 |
| | GT-GAN | 0.90 | **1.00** | **1.00** | 0.94 |
| | optimal | 1.00 | 1.00 | 1.00 | 1.00 |
| | GraphRNN | 0.95 | 0.99 | **1.00** | 0.96 |
| 150 | S-Generator | 0.50 | **1.00** | 0.49 | 0.66 |
| | GT-GAN | **0.97** | **1.00** | **1.00** | **0.98** |
| | Gold Standard | 1.00 | 0.99 | 1.00 | 0.99 |

Table 4: Indirect evaluation in user authentication graph datasets

| Graph size | Method | Precision | Recall | AUC | F1 |
|---|---|---|---|---|---|
| | RandomVAE | 0.32 | 0.51 | 0.26 | 0.39 |
| | GraphRNN | 0.34 | 0.36 | 0.50 | 0.36 |
| 50 | S-Generator | 0.72 | 0.61 | 0.74 | 0.66 |
| | GT-GAN | **0.79** | **0.68** | **0.78** | **0.73** |
| | Gold Standard | 0.97 | 0.97 | 0.97 | 0.97 |
| | S-Generator | 0.77 | 0.58 | 0.62 | 0.66 |
| 300 | GT-GAN | **0.84** | **0.66** | **0.79** | **0.74** |
| | Gold Standard | 0.98 | 0.96 | 0.97 | 0.97 |

**Results on User Authentication Graphs**: As shown in Table 4, classifiers trained by the graphs generated by GT-GAN can effectively classify normal and hacked behaviors with AUC above 0.78, largely above the 0.5 if using random model. GT-GAN significantly outperforms other methods by around 25%, 16%, 24.5% and 22.1% respectively on four metrics. GT-GAN performs consistently better than other baselines when the graph size varies from 50 to 300. More indirected evaluation results can be found in Fig. 9 and Fig. 8 in Appendix C, including the case studies.

## 5 CONCLUSION AND FUTURE WORKS

This paper focuses on a new problem on deep graph translation. To achieve this, we propose a novel Graph-Translation-Generative Adversarial Networks (GT-GAN) which will generate a graph translator from input to target graphs. To learn both the global and local graph translation mapping for the directed weighted graph, new graph convolutions and deconvolutions have been proposed to encode and decode the graphs while preserving the high-level and local-level graph patterns. Extensive experiments have been done on the synthetic and real-world dataset with the comparison with the state-of-the-art method in graph generation. Experiments show that our GT-GAN indeed can discover the ground-truth translation rules, and significantly outperform the comparison method in both effectiveness and scalability. This paper opens a new window for deep graph translation. There are many possible future directions yet to explore, such as develop domain-specific graph translator using domain knowledge or analyze and visualize the deep translation patterns.

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

# A MORE EXPERIMENTAL RESULTS FOR SCALE FREE GRAPH SET

Fig. 6 shows 18 examples for scale free dataset from size 50 to 150.

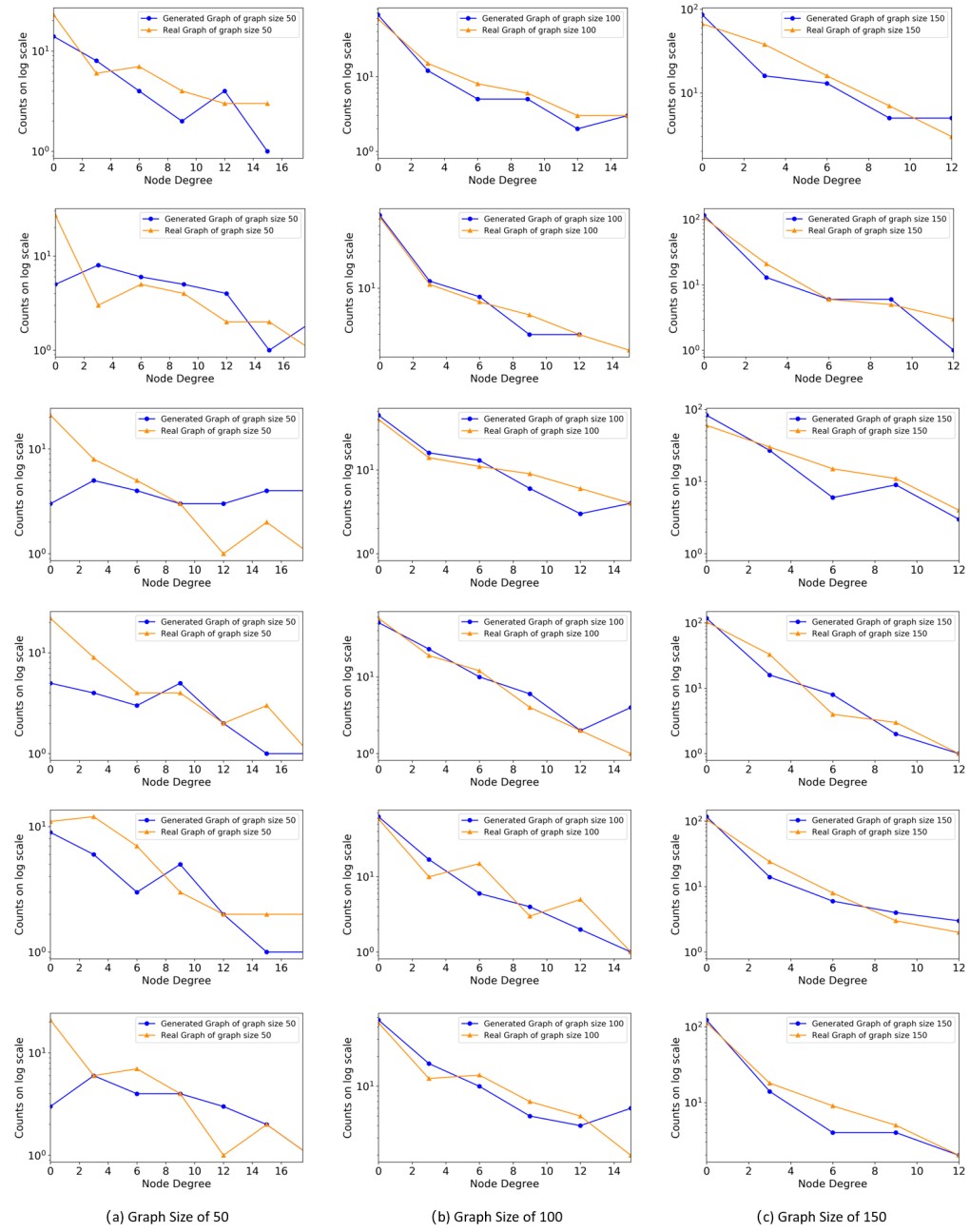

Figure 6: Examples of node degree distrbution for generated graphs and real graphs

# B MORE EXPERIMENTAL RESULTS FOR POISSON RANDOM GRAPH SET

For Poisson random graphs, the distributions of $k$ in the real target graphs and those generated graphs are compared. The mean of edge increasing ratio $k$ for generated graphs by our GT-GAN is 3.6, compared to the real value of 5, which implies that the GT-GAN generally is able to discover the underlying increasing ratio between input and target graphs. More evaluation results (e.g. degree

and repository) can be found in Appendix B. We draw the probability density curve of the proportion k. Fig. 7 shows the distribution of the $k$ in graphs generated by GT-GAN and the real graphs. The distribution plot is drew based on 3000 samples. Both of the two distribution have main degree values in the range from 2 to 7, while there is difference in the max frequency due to the limit of the samples amount. However, it prove that the proposed GT-GAN do learn the distribution type of translation parameter $k$ in this task.

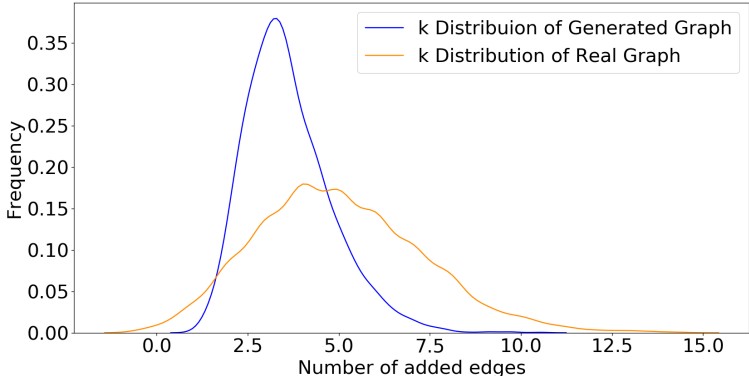

Figure 7: Distribution of $k$ for generated graphs and real graphs in Poisson random graph set

Table 5 shows the distance measurement between generated graphs and real graphs in several metrics. For the metric "degree", we use Wasserstein distances to measure the distance of two degree distribution. For other metrics, we calculate the MSE between generated graphs and real graphs.

Table 5: MSE of Graph properties measurements for poisson random graphs

| Graph size | Method | Density | Average Degree | Reciprocity |
|---|---|---|---|---|
| 10 | RandomVAE | 0.1772 | 2.8172 | 0.3917 |
| | GraphRNN | 0.2665 | 2.2078 | 0.1344 |
| | GrapgGMG | 0.3519 | 2.4286 | 0.1338 |
| | GraphVAE | 0.2881 | 3.1986 | 0.3103 |
| | S-Generator | **0.2993** | **1.575**1 | **0.0737** |
| | GT-GAN | 0.3084 | 1.7707 | 0.1327 |
| 20 | RandomVAE | 0.2078 | 7.0860 | 0.4182 |
| | GraphRNN | 0.2305 | 4.9256 | 0.1190 |
| | S-Generator | 0.2111 | 3.2207 | 0.0430 |
| | GT-GAN | **0.2013** | **3.2047** | **0.0388** |
| 50 | RandomVAE | Inf | 23.680 | 0.5362 |
| | GraphRNN | **0.0110** | 3.6000 | 0.0125 |
| | S-Generator | 0.0120 | **2.9082** | 0.0125 |
| | GT-GAN | 0.0155 | 3.2960 | **0.0047** |
| 100 | GraphRNN | 0.0123 | 3.5475 | 0.0034 |
| | S-Generator | **0.0029** | **2.9167** | **0.0034** |
| | GT-GAN | 0.0142 | 4.3730 | 0.0043 |
| 150 | GraphRNN | **0.0012** | 3.6619 | 0.0016 |
| | S-Generator | 0.0013 | **2.9467** | **0.0016** |
| | GT-GAN | 0.0061 | 5.0410 | 0.0019 |

Table 6: MSE of Graph properties measurements for user authentication dataset

| Graph size | Method | Density | Reciprocity | Average Degree |
|---|---|---|---|---|
| 50 | RandomVAE | 0.0005 | 0.0000 | 6.4064 |
| | GraphRNN | 0.0032 | 0.0000 | 2.7751 |
| | S-Generator | 0.0244 | 0.0342 | 24.130 |
| | GT-GAN | **0.0003** | **0.0000** | **0.0002** |
| 300 | S-Generator | 0.0113 | 0.0010 | 8.6839 |
| | GT-GAN | **0.0004** | **0.0000** | **0.0006** |

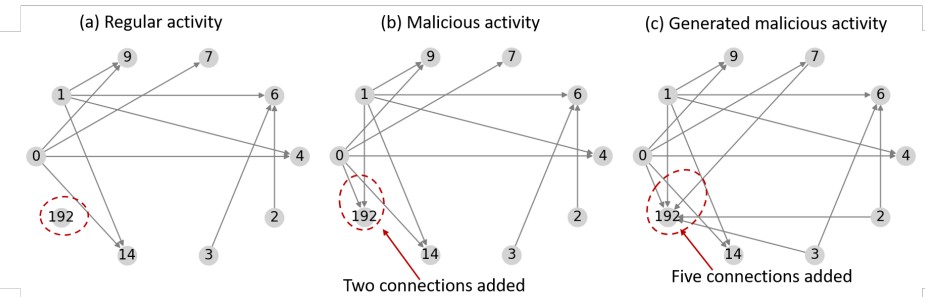

Figure 8: Regular graphs, malicious graphs and generated graphs of User 049

## C MORE EXPERIMENTAL RESULTS FOR USER AUTHENTICATION GRAPH SET

**About Original Dataset**   This data set spans one calendar year of contiguous activity spanning 2012 and 2013. It originated from 33.9 billion raw event logs (1.4 terabytes compressed) collected across the LANL enterprise network of approximately 24,000 computers. Here we consider two sub dataset. First is the user log-on activity set. This data represents authentication events collected from individual Windows-based desktop computers, servers, and Active Directory servers. Another dataset presents specific events taken from the authentication data that present known red team compromise events, as we call malicious event. The red team data can used as ground truth of bad behavior which is different from normal user. Each graph can represent the log-on activity of one user in a time window. The event graphs are defined like this: The node refers to the computers that are available to a user and the edge represents the log-on activity from one computer to another computer of the user.

**Direct evaluation of User authentication Graph Set**   For the user authentication graphs, the real target graphs and those generated are compared under well-recognized graph metrics including degree of nodes, reciprocity, and density. We calculate the distance of degree distribution and Mean Sqaured Error (MSE) for reciprocity and density. Results can be found in Appendix C. We use seven metrics to evaluate the similarity of generated graphs and real graphs. The MSE value are calculated to measure the similarity between two graphs in term of different metrics. 6 shows the mean square error of the generated graphs and real graphs for all users.

**Case Studies on the generated target graphs**   Fig. 8 shows the example of User 049 with regular activity graph, real malicious activity graph and malicious activity graph generated by our GT-GAN from left to right. Only those of edges with difference among them are drawn for legibility. It can be seen that, the hacker performed attacks on Computer 192, which has been successfully simulated by our GT-GAN. In addition, GT-GAN also correctly identified that the Computer 192 is the end node (i.e., with only incoming edges) in this attack. This is because GT-GAN can learn both the global hacking patterns (i.e., graph density, modularity) but also can learn local properties for specific nodes (i.e., computers). GT-GAN even successfully predicted that the hacker connect from Computers 0 and 1, with Computers 7 and 14 as false alarms.

For User006 in Fig. 9, the red team attackers make more connections on Node 36 compared to user's regular activity, as marked in red rectangle. GT-GAN leans how to choose the Node 36 and it generated more connections too in the Node 36 .

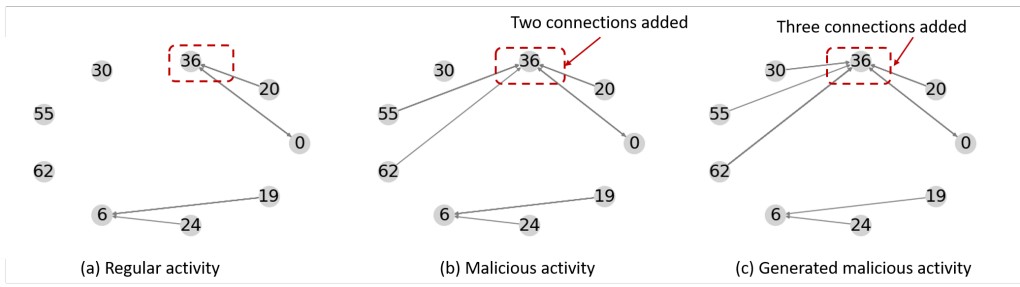

Figure 9: Regular graphs, malicious graphs and generated graphs for User 006

## D    FLOWCHART OF INDIRECT EVALUATION PROCESS

Fig. 10 shows the process of the indirect evaluation process.

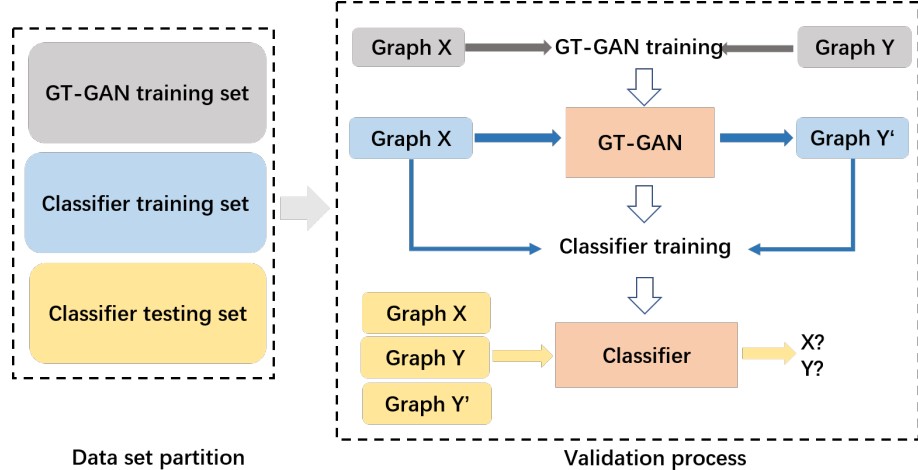

Figure 10: Flow chart of validation

## E    ARCHITECTURE PARAMETER FOR GT-GAN MODEL

**Graph Generator**: Given the graph size (number of nodes) $N$ of a graph. The output feature map size of each layer through graph generator can be expressed as:

$$N \times N \times 1 \to N \times N \times 5 \to N \times N \times 10 \to N \times 1 \times 10 \to N \times N \times 10 \to N \times N \times 5 \to N \times N \times 1$$

**Discriminator**: Given the graph size (number of nodes) $N$ of a graph. The output feature map size of each layer through graph discriminator can be expressed as:

$$N \times N \times 1 \to N \times N \times 5 \to N \times N \times 10 \to N \times 1 \times 10 \to 1 \times 1 \times 10$$

For the edge to edge layers, the size of two kernels in two directions are $N \times 1$ and $1 \times N$. For the node to edge layer, the kernel size is $1 \times N$

