# OpenReview forum: "DEEP GRAPH TRANSLATION"
_ICLR.cc/2019/Conference_

### Official Review · AnonReviewer2 · 2018-11-01
**Interesting work with some odd issues on implementation and results**

**Rating:** 6
**Confidence:** 4

**Review:**

The paper presents an approach for translating graphs in one domain to graphs in the same domain using a GAN approach. A graph Translator approach is defined and a number of synthetic data sets and one real-world data set are used to evaluate the approach. Most of the paper is written well, though there are some odd sentence structure issues in places. The paper could do with a thorough check for grammatical and spelling mistakes. For example you miss-spell NVIDIA.

The main concerns with the work:
1) Equation 2 is used to minimise the distance between graphs from X and graphs in Y. Yet, the main metric which is used to evaluate the paper is this distance. This would seem to give an unfair advantage to your approach. I would also be concerned about the fact that later you use this for stating if a graph represents good or hacker activity. If you have drawn translated graphs towards real graphs, how do you know that you haven’t pulled a good graph closer to a hacker graph? This is more concerning considering work which came out of NIPS which suggested that GAN’s tend to favour producing similar output rather than spreading it evenly over the domain.

2) It isn’t entirely clear what your results are trying to show. Presumably P, R, AUC and F1 are generated from the results produced from your Discriminator? Were each of the other approaches optimised against your discriminator or not? Also, it is unclear as to what the Gold Standard method is - we’re only told that its a classifier, but what type and how constructed?

3) Your approach seems to be ‘fixed’ in the set of nodes which are in both in the input and output graphs - needing to be the same. This would seem significantly limiting as graphs are rarely of the same node set.

4) Although you comment on other graphs approaches being limited to very small graphs, you do not test your approach on graphs with over 150 nodes. These would also seem to be very small graphs in comparison to real-world graphs. Further evaluation on larger graphs would seem to be essential - how long would it take on graphs with 10^6 nodes?

5) The real-world dataset seems rather odd and not fully explored. Given that you have this data it is surprising that you didn’t complete the loop by showing that you could take data from before a hack attempt and show that you could predict that in the future you had a hack attempt. Perhaps this is due to the fact that you didn’t have the ground-truth data in here to show a graph going from good to bad? But if not it would have been good to have shown, either through this data or some other, how your approach does match in with real-world results.

Given the points above, I would be very concerned on an approach which used the above to identify a future hacking attempt.

Some more specific comments on the paper:
- "The tremendous success of deep generative models on generating continuous data like image and audio” - it is not clear what this continuous data is.

- Hard to parse : “which barely can be available for the accounts worth being monitored.”

- “This requires us to learn the generic distribution of theft behaviors from historical attacks and synthesize the possible malicious authentication graphs for the other accounts conditioning on their current computer networks” - given that these historical attacks are (hopefully) rare, is there enough data here to construct a model?

- Please define GCNN

- “Our GT-GAN is highly extensible where underlying building blocks, GCNN and distance measure in discriminator, can be replaced by other techniques such as (Kipf & Welling, 2017; Arjovsky et al., 2017) or their extensions.” - this sounds more like a feature of what you have contributed rather than a contribution in its own right.

- In the context of synthetic data, what is ground-truth?

- Hard to parse “Modern deep learning techniques operating on graphs is a new trending topic in recent years.”

- Hard to parse “However, these methods are highly tailored to only address the graph generation in a specific type of applications such as molecules generation”

- Hard to parse “Existing works are basically all proposed in the most recent year,”

- “Typically we focus on learning the translation from one topological patterns to the other one” -> “Typically we focus on learning the translation from one topological pattern to the other”

- It’s not clear in equation 1 how you represent G_X. Only much later is it mentioned about adjacency matrix.

- Hard to parse “Different and more difficult than graph generation designed only for learning the distribution of graph representations, for graph translation one needs to learn not only the latent graph presentation but also the generic translation mapping from input graph to the target graph simultaneously.“

- Hard to parse “graph translation requires to learn”

- Hard to parse “in most of them the input signal is given over node with a static set of edge and their weights fixed for all samples”

- “we propose an graph” -> “we propose a graph”

- Hard to parse “The two components of the formula refers to direction filters as talked above”

- Hard to parse “Next, graph translator requires to”

- “as shown in Equations equation 7 and Equations equation 6,” -> “as shown in Equation 6 and Equation 7”

- Hard to parse “The challenge is that we need not only to learn the”

- Figure 2 would seem to need more explanation.

- The end of section 3.3 is a bit vague and lacks enough detail to reproduce.

- “our GT-GAN is able to provide a scalable (i.e., O(N2)) algorithm that can generate general graphs.” - what sizes have you tested this up to?

- Hard to parse “we randomly add another kjEj edges on it to form the target graph”

- “The goal is to forecast and synthesize the future potential malicious authentication graphs of the users without any historical malicious behaviors, by the graph translator from normal to malicious graph trained based on the users with historical malicious-behavior records.” - This isn’t entirely clear. Are you trying to create new malicious graphs or show that a current graph will eventually go malicious?

- “All the comparison methods are directly trained by the malicious graphs without the conditions of input graphs as they can only do graph generation instead of translation.” - not clear. For the synthetic data sets how did you choose which ones were malicious?

- “GraphRNN is tested with graph size within 150. GraphGMG, GraphVAE is tested within size 10 and RandomVAE is tested on graphs within size 150.” -> “GraphRNN and RandomVAE are tested with graph up to size 150. GraphGMG, GraphVAE is tested with graphs up to  size 10.”

- “Here, beyond label imbalance, we are interested in “label missing” which is more challenging.” - “missing labels”?

- “In addition, we have also trained a “gold standard” classifier based on input graphs and real target
graphs.” - need to say more about this.

---

> ### Author Response · Authors · 2018-11-12
> **Explanations for concerns：Part I**
>
>
> Dear Reviewer:
>
> Thanks very much for your comments and questions. We would like to first explain your concerns.
>
> -------------------------
> Q: Equation 2 is used to minimize the distance between graphs from X and graphs in Y. Yet, the main metric which is used to evaluate the paper is this distance. This would seem to give an unfair advantage to your approach. I would also be concerned about the fact that later you use this for stating if a graph represents good or hacker activity. If you have drawn translated graphs towards real graphs, how do you know that you haven’t pulled a good graph closer to a hacker graph? This is more concerning considering work which came out of NIPS which suggested that GAN’s tend to favour producing similar output rather than spreading it evenly over the domain.
>
> A: (1) We minimize the distance between generated graphs and real target graphs in Y, not graphs from X and graphs from Y. The comparison methods also minimize the distance between their generated graphs and real graphs, which are the same as us.
> (2) We have done both direct and indirect evaluations in all the datasets, which comprehensively demonstrate the good performance of the proposed methods. Moreover, the indirect evaluations (see Table 2-4) do not use distance to evaluate the performance but the classification metrics (precision, recall, ACU and F1-score). Even in direct evaluation, the metrics contain degree distribution distance, MSE of adjacent matrix comparison and repository comparing, while the loss function in equation 2 is applied only on the MSE of the adjacent matrix of graphs.
>
> For the statement “NIPS which suggested that GAN’s tend to favor producing similar output”. Our answers are three-fold:
> (1) As we have done an extensive survey and did not find papers using GAN for graph generation yet, we doubt if the experience in GAN on other data still applies exactly the same for graph data. This is because as we know, graphs are highly different types of data than images which are continuous-valued (e.g., RGB). In graphs, nodes can have arbitrary connectivity.
> (2) Moreover, our experiments suggest that there is indeed nontrivial variance in the generated graphs. As shown in Figure 5 in Appendix, we can see the difference among the degree distributions of different graphs we generated is obvious.
> (3) We can tune the dropout ratio to control the degree of variation of generated output. The noise is introduced by the dropout function in each convolution layer, which functions by randomly ignoring certain ratio of neuron’s output of a network.
>
> -------------------------
> Q: It isn’t entirely clear what your results are trying to show. Presumably P, R, AUC and F1 are generated from the results produced from your Discriminator? Were each of the other approaches optimized against your discriminator or not? Also, it is unclear as to what the Gold Standard method is - we’re only told that it’s a classifier, but what type and how constructed?
>
> A: (1) P, R, AUC and F1 are metrics to indirectly evaluate the generated graphs by comparison models.  Since all the comparison methods in our paper are generative models which generate graphs, and hence our experiment is to evaluate how good the generated graphs are. One way to evaluate this is by “indirect evaluation”, where we use the graphs generated by different comparison methods as training data to train a classifier (all are based on KCNN for fairness), and then compare which classifier is better. The flowchart of the indirect evaluation is shown in Figure 9, in Appendix D.
> (2) “Gold standard” is the classifier trained directly by the real target graphs, instead of generated graphs. As it directly uses the real graphs to train the classifier (still based on KCNN), it is expected to get the best performance. Therefore, “gold standard” method acts as the “best-possible-performer”, and is used as a benchmark to evaluate all the different generative models on how “real” the graphs they can generate: the closer (and better) their performance is to the “gold standard” one, the “more real” their generated graphs are.
>
> -------------------------
> Q: Your approach seems to be ‘fixed’ in the set of nodes which are in both in the input and output graphs - needing to be the same. This would seem significantly limiting as graphs are rarely of the same node set.
>
> A: Yes, we admit that our model has a limitation in dealing with the variable-size input graphs. This limitation largely exists in the existing deep graph learning methods, especially those based on graph convolution. This problem itself is a challenging open problem that requires significant future efforts in the community. However, the focus of our work in this paper is the translation mapping establishment, optimization, and evaluation. We are indeed considering one of our next extensions to deal with this problem. Thanks for the comments.

---

> > ### Comment · AnonReviewer2 · 2018-12-05
> > **Wow, that's a lot of replies**
> >
> > Thanks for the comments. I still think the work is interesting and the comments and improvements to the paper help. I'm unfortunately not convinced that it's yet good enough to go up to the next category.

---

> ### Author Response · Authors · 2018-11-12
> **Explanations for concerns：Part II**
>
>
> -------------------------
> Q: Although you comment on other graphs approaches being limited to very small graphs, you do not test your approach on graphs with over 150 nodes. These would also seem to be very small graphs in comparison to real-world graphs. Further evaluation on larger graphs would seem to be essential - how long would it take on graphs with 10^6 nodes?
>
> A: (1) Our testing experiments do test graphs on size 300 (i.e., user authentication dataset), but is not yet able to handle the scale of millions of nodes. Handling millions of nodes in graph is much more difficult than handling millions of pixels in images. This is because in graph data the nodes connect arbitrarily, and the adjacency matrix is required to code the connectivity among nodes which are at least quadratic to the number of nodes. And this is partially why most of the existing work on graph generative learning domain focus on only a very small graph. And also because of this, we test small-size graphs in this paper because most of the comparison methods can typically only handle dozens of nodes or fewer graphs. Compared to them, our model handles nontrivially “larger graph” (6-10 times larger than most existing methods).
> (2) Many important real-world applications are small graphs and may need graph generation. To list a few (with graph size from dozens to thousands): chemical molecules, cyber network (in our paper), electrical circuits, and semantic network (where nodes are words and links are their semantic correlations).
>
> -------------------------
> Q: The real-world dataset seems rather odd and not fully explored. Given that you have this data it is surprising that you didn’t complete the loop by showing that you could take data from before a hack attempt and show that you could predict that in the future you had a hack attempt. Perhaps this is due to the fact that you didn’t have the ground-truth data in here to show a graph going from good to bad? But if not, it would have been good to have shown, either through this data or some other, how your approach does match in with real-world results.
>
> A: (1) The real-world dataset and its application are authoritative and motivated this research. The dataset is recent, authoritative, and provided by the prestigious “Los Alamos National Laboratory” (https://csr.lanl.gov/data/cyber1/ ). The research problem behind this dataset raised up their needs to predict the future hacking behavior of a user with no historical hacking behavior has been a highly practical but prohibitively challenging. Such application strong motivates this new domain of graph translation where we transfer the hacking behavior from those users with historical hacking behavior in different network structure.
> (2) The dataset has been fully explored by a loop. We indeed have predicted the hack attempt in the future and validated it against ground truth with accuracy metrics such as accuracy, F1, Precision, and recall explained in the last paragraph of Section 4.2.3. Specifically,  we use half of users’ good graphs (data before attack) and real hacker graphs (data after attack) to train the translator. We then generate graphs with this translator for the other users. To evaluate indirectly, we use the generated graphs and good graphs to train an attacker prediction model (classifier) for each user, if it can recognize his real hacker graphs, this prediction model works in real-world and our translator is good.
> (3) We have also shown the case studies on “a graph going from good to bad”. Specifically, in Figures 7 and 8 in Appendix C, we have shown: 1) the “good” graph, 2)the “hacked” graph generated by our methods, and 3) the ground-truth “hacked” graph. And the results show that our methods can well predict the hack attempts. During our experiments, we have observed numerous such case studies and put them as representatives.

---

> ### Author Response · Authors · 2018-11-12
> **Some statement explanations and modifications: Part I**
>
> Next, we would like to reply to more specific comments.
>
> ----------------------
> Q: "The tremendous success of deep generative models on generating continuous data like image and audio” - it is not clear what this continuous data is.
>
> A：The continuous data here means that the elements (pixel or signal) in an image or audio is continuous in Euclidean structure and their features are fully related to spatial positions of elements.
>
> ----------------------
> Q: “This requires us to learn the generic distribution of theft behaviors from historical attacks and synthesize the possible malicious authentication graphs for the other accounts conditioning on their current computer networks” - given that these historical attacks are (hopefully) rare, is there enough data here to construct a model?
>
> A: (1) Yes, we have enough data here to train a good model since we used the historical attack data from all users, instead of constructing a model for each individual user. There are 315 training samples for the user authentication dataset, which is sufficient to get the good performance as shown in Tables 4 in Section 4.2.3  and Table 6 in Appendix C.
> (2) One attacker’s rule is supposed to be shared among all the users. Thus, once parts of a company are attacked by one attacker, we can train a translator model by the data from the attacked users and generate synthetic attack activities for the un-attacked users.
>
> ----------------------
> Q: Please define GCNN
>
> A: Sorry for this error, we have added the definition in the revised version. GCNN is short for Graph Convolution Neural Network, which is a general term referring to the encoder model in our framework.
>
> ----------------------
> Q: “Our GT-GAN is highly extensible where underlying building blocks, GCNN and distance measure in discriminator, can be replaced by other techniques such as (Kipf & Welling, 2017; Arjovsky et al., 2017) or their extensions.” - this sounds more like a feature of what you have contributed rather than a contribution.
>
> A: we would like to state that that DGT is more like a general graph translation framework than a special model. The encoder and decoder model we use here can be regarded as a special case and it can also be replaced by others with different distance measurements.
>
> ----------------------
> Q: In the context of synthetic data, what is ground-truth?
>
> A: (1) In the synthetic dataset, there are also input graphs and target graphs, just like the real-world dataset.
> (2) Target graphs are generated based on the input graphs by following some predefined rules. We use the input graphs and target graphs to train the translator. Thus, to evaluate the generated graphs, the target graphs are the ground-truth. The detail of the generation rules of target graphs is given in 4.1.1 Datasets.
>
> ----------------------
> Q: Figure 2 would seem to need more explanation.
>
> A: (1) Figure 2 shows the difference between the graph convolution and graph deconvolution, which is stated in Section 3.2.2 Graph deconvolutions. The encoder does n-hop edge information aggregation from the input graphs and learns the latent representation of nodes. Then we first decode the node embedding to get the n-hop aggregated information on edges by node-to-edge layer and then we further decode the n-hop aggregated information by n-layers back to get the output adjacency matrix.
> (2) we have modified our paper in Section 3.2.2, e.g, by adding “To get the nth hop information Aij, row filter decodes all the (n+1)-th hop information of outgoing edges of Vi and column filter decodes all the (n+1)-th hop information of incoming edges of Vj.”
>
>  ----------------------
> Q: The end of section 3.3 is a bit vague and lacks enough detail to reproduce.
>
> A: (1) Function of discriminator: In the training phase, the discriminator aims to classify the generated graphs and the real target graphs. The translator and discriminator are trained together, and the final goal is that the discriminator cannot distinguish the generated graphs and real target graphs.
> (2) Construction: The discriminator is constructed by an edge-to-edge layer and edge-to-node layer, which learns features from graphs like the translator. Then the extracted features of each node are together mapped into a fully connected layer, embedding the graph into a vector which is inputted into a SoftMax classifier to output the probability of the graph to be a target graph or generated graph. We have modified this section for better readability. We have released our codes in the paper and it will give an additional guarantee that the experimental results can be easily reproduced.
>
> ----------------------
> Q: “our GT-GAN is able to provide a scalable (i.e., O(N2)) algorithm that can generate general graphs.” - what sizes have you tested this up to?
>
> A: We have test size up to 300.

---

> ### Author Response · Authors · 2018-11-12
> **Some statement explanations and modifications: Part II**
>
>
> ----------------------
> Q: “The goal is to forecast and synthesize the future potential malicious authentication graphs of the users without any historical malicious behaviors, by the graph translator from normal to malicious graph trained based on the users with historical malicious-behavior records.” - This isn’t entirely clear. Are you trying to create new malicious graphs or show that a current graph will eventually go malicious?
>
> A: (1) What we do: For a user who has not been attacked, we want to synthesize what would be the attack activities like if he was attacked. To do this, we train a translator from the other user's historical data and use this translator to generate attacked activities for this user.
> (2) Why we do: If we can generate the attacked graphs of a user, we can build a prediction model (classifier) for this user before he is really attacked.
>
>
> ----------------------
> Q: “All the comparison methods are directly trained by the malicious graphs without the conditions of input graphs as they can only do graph generation instead of translation.” - not clear. For the synthetic data sets how did you choose which ones were malicious?
>
> A: Sorry for this statement error, it should be modified as: “All the comparison methods are directly trained by the target graphs without the conditions of input graphs as they can only do graph generation instead of translation.”
>
> ----------------------
> Q: “GraphRNN is tested with graph size within 150. GraphGMG, GraphVAE is tested within size 10 and RandomVAE is tested on graphs within size 150.” -> “GraphRNN and RandomVAE are tested with graph up to size 150. GraphGMG, GraphVAE is tested with graphs up to size 10.”
> A: We have modified it as: “RandomVAE is tested on graphs within size 50.”
>
> ----------------------
> Q: “Here, beyond label imbalance, we are interested in “label missing” which is more challenging.” - “missing labels”?
> A: Sorry for the confusion. We take an example for Missing label in the authentication attack data. In a company, only some groups of users have been attacked and thus these users can easily build a prediction model based on both positive and negative data. In contrast, some other users who have not been attacked will not have any positive examples, which we called the “label missing” problem.
>
>  ----------------------
> Q: “In addition, we have also trained a “gold standard” classifier based on input graphs and real target graphs.” - need to say more about this.
> A: Yes, we have explained the gold standard in above answers and more detailed explanation can be found in 1st and 2nd paragraphs of 4.2.3.
>
> ----------------------
> Q: Hard to parse: “which barely can be available for the accounts worth being monitored.”
> A: We have modified it as: “which barely can be available since these accounts always being monitored well”.
>
> ----------------------
> Q: Hard to parse “Modern deep learning techniques operating on graphs is a new trending topic in recent years.”
> A: We have modified it as: “Deep learning techniques dealing with graphs is a new trending topic in recent years.”
>
> ----------------------
> Q: Hard to parse “However, these methods are highly tailored to only address the graph generation in a specific type of applications such as molecules generation”
> A: We have modified it as: “However, these methods are not general for being widely applied”
>
> ----------------------
> Q: Hard to parse “Existing works are basically all proposed in the most recent year,”
> A: We have modified it as: “Existing works are almost proposed in the most recent year,”
>
> ----------------------
> Q: “Typically, we focus on learning the translation from one topological pattern to the other one” -> “Typically we focus on learning the translation from one topological pattern to the other”
> A: Thanks for correctness. We have modified in revised paper.
>
> ----------------------
> Q: It’s not clear in equation 1 how you represent G_X. Only much later is it mentioned about adjacency matrix.
> A: Yes, the graphs here are represented as the weighted adjacent matrix of a graph.

---

> ### Author Response · Authors · 2018-11-12
> **Some statement explanations and modifications: Part III**
>
> ----------------------
> Q: Hard to parse “Different and more difficult than graph generation designed only for learning the distribution of graph representations, graph translation one needs to learn not only the latent graph presentation but also the generic translation mapping from input graph to the target graph simultaneously. “
> A: We have modified it as: “Different and more difficult than graph generation designed only for learning the distribution of graph representations, graph translation learns not only the latent graph presentation but also the generic translation mapping from input graph to the target graph simultaneously. “
>
> ----------------------
> Q: Hard to parse “graph translation requires to learn”
> A: We have modified it as: “graph translation focus on learning….”
>
> ----------------------
> Q: Hard to parse “in most of them the input signal is given over node with a static set of edge and their weights fixed for all samples”
> A: We have modified it as: “most of them try to embed the graphs based on the signals assigned on the nodes with the fixed graph topology”
>
> ----------------------
> Q: “we propose an graph” -> “we propose a graph”
> A: Thanks for correctness. We have modified in revised paper.
>
> ----------------------
> Q: Hard to parse “The two components of the formula refers to direction filters as talked above”
> A: We have modified it as: “The two components refers to out-direction and in-direction features as talked above”
>
> ----------------------
> Q: Hard to parse “Next, graph translator requires to”
> A: We have modified it as: “Next, decoder part aims to”
>
> ----------------------
> Q: “as shown in Equations equation 7 and Equations equation 6,” -> “as shown in Equation 6 and Equation 7”
> A: Thanks for correctness. We have modified in revised paper.
>
> ----------------------
> Q: Hard to parse “The challenge is that we need not only to learn the”
> A: We have modified it as: “The challenge is not only learning the”
>
> ----------------------
> Q: Hard to parse “we randomly add another kjEj edges on it to form the target graph”
> A: We have modified it as: “we randomly add kjEj edges on it to form the target graph”
>
> We hope we were able to explain everything clearly to your satisfaction, please let us know if there are any more open points.
>
> Thank you once again!

---

### Official Review · AnonReviewer3 · 2018-11-02
**Good problem setting, interesting results, needs more clarifications.**

**Rating:** 5
**Confidence:** 4

**Review:**

This paper addresses the important / open problem of graph generation, and specifically in a conditional/transductive setting.

Graph generations is a new topic, it is difficult, and has many important applications, for instance generating new molecules for drug development.

As stated by the authors, this is a relatively open field: there are not many papers in this area, with most approaches today resorting to domain specific encodinings, or "flattening" of graphs into sequences to then allow for the use recurrence (like in MT); this which per se is an rather coarse approximation to graph topology representations, thus fully motivating the need for new solutions that take graph-structure into account.

The setting / application of this method to graph synthesis of suspicious behaviours of network users, to detect intrusion, effectively a Zero-shot problem, is super interesting.

The main architectural contribution of this paper are graph-deconvolutions, practically a graph-equivalent of CNN's depth-to-space - achieved by means of transposed structural matrix multiplication of the hidden GNN (graph-NN) activation - simple, reasonable and effective.

While better than most of the baseline methods, the N^2 memory/computational complexity is not bad, but still too high to scale to very large graphs.

Results are provided on relatively new tasks so it's hard to compare fully to previous methods, but the authors do make an attempt to provide comparisons on synthetic graphs and intrusion detection data. The authors do published their code on GitHub with a link to the datasets as well.

As previously mentioned in public comments on this forum, some points in the paper are not very clear; specifically regarding the loss function, the definition of "edge-to-edge" convolutions and generally the architectural choice related to the conditional GAN discriminator. Clarifications of these points, and more in general the philosophy behind the architectural choices made, would make this paper a much clearer accept.

Thank you!

ps // next my previous public comments, in detail, repeated ...

--

- the general architecture, and specifically the logic behind the edge-to-edge convolution, and generally the different blocks in fig.1 "graph translator".

- how exactly do you do a L1 loss on graphs? I'd have to assume the topology of the graph is unchanged between Gy and T(Gx) ~ and then maybe take L1 of weight matrix? But then is this general enough ~ given your stated goal of modeling different topologies? Either ways, more explanation / and perhaps equations to clarify this loss would be very helpful.

- why do you need a conditional GAN discriminator, if you already model similarity by L1? Typically one would use a GAN-D() to model "proximity" to the source-distribution, and then a similarity loss (L1 in your case) to model "proximity" to the actual input sample, in the case of trasductional domains. Instead here you seem to suggest to use L1 and GAN to do basically the same thing, or with significant overlap anyways. This is confusing to me. Please explain the logic for this architectural choice.

-  could you please explain the setting for the “gold standard” experiment. I'd have to assume, for instance, you train a GNN in a supervised way by using both source (non-suspicious) and target (suspicious) behaviour, and label accordingly? That said I am not 100% sure of this problem setting.

---

> ### Author Response · Authors · 2018-11-10
> **Clarifications of confused points, and the philosophy behind the architecture: Part I**
>
> Dear Reviewer:
>
> Thanks very much for your comments and questions. We would like to explain them in detail and modify our paper accordingly.
>
> ----------------------------------------------------------------------------
> Q: First, the general architecture, and specifically the logic behind the edge-to-edge convolution, and generally the different blocks in fig.1 "graph translator".
>
> A: General architecture: The whole framework includes a translator and a discriminator.
> (1) Translator. Translator consists of an encoder, a decoder, and a skip network, which first learn the representation of the graph and then decode it back to the target graph. See details in the third part of the answer.
> (2) Discriminator. Our discriminator aims to classify the generated graphs and the real target graphs given the input graph.
> (3) The translator and discriminator are trained together, and the final goal is that the discriminator cannot distinguish the generated graphs and real target graphs. After training such a model, the translator will be used in the test phase.
>
> The logic behind edge-to-edge convolution:
> (1) Generally speaking, the purpose of edge-to-edge convolution layers is to aggregate the neighborhood information of nodes. Specifically, the n-th edge-to-edge convolution layer aggregates the n-th hop connection information of nodes related to each edge.
> (2) Different from image convolution, for each hidden channel, we have two filters, one is a column vector while the other is a row vector. To learn the nth hop information of edge <i,j>, row filter aggregates all the (n-1)-th hop information of outgoing edges of node i and column filter aggregates all the (n-1)-th hop information of incoming edges of node j.
> (3)  Edge-to-edge layers are important to extract some higher-level graph features, e.g., the n-hop reachability from a node to another; n-hop in-degree and out-degree, and many other higher-order patterns.
>
> Different blocks in the graph translator:
> Translator consists of an encoder, a decoder, and a skip network.
> (1) Encoder. The encoder does n-hop edge information aggregation from the input graphs using edge-to-edge layers and then uses the edge-to-node layer to learn the latent representation of nodes.
> (2) Decoder. Reversely, the graph decoder first uses node-to-edge layers to decode the node representations to aggregated edge information and then further decode that into adjacency matrix, which is the final generated graphs.
> (3) Skip-network. Over the encoder-decoder framework, we also added skip-network (the black line of Fig.1) which can directly map the edge aggregation information in every hop from the input graph to the output graph so that can preserve the local information in every resolution (i.e., every hop).
>
> ----------------------------------------------------------------------------
> Q: how exactly do you do a L1 loss on graphs? I'd have to assume the topology of the graph is unchanged between Gy and T(Gx) ~ and then maybe take L1 of weight matrix? But then is this general enough ~ given your stated goal of modeling different topologies? Either ways, more explanation / and perhaps equations to clarify this loss would be very helpful.
>
> A: (1) L1 norm is applied to the weight matrix. Our methodology is still general enough which is achieved by a trade-off between L1 loss and adversarial loss (GAN-D), which jointly enforces Gy and T(Gx) to follow a similar topological pattern but may not necessarily the same. Specifically, L1 makes T(Gx) share the same rough outline of sparsity pattern like Gy, while under this outline, adversarial loss allows the T(Gx) to vary to some degree.
> (2) Combining L1 loss and adversarial loss is well-recognized and validated. Works on image-translation have proposed and utilized L1 loss and adversarial loss jointly in GAN, for example, reference [1] (with 600+ citations) and reference [2] (with 1300+ citations). They have done extensive experiments to show the advantage of such a strategy. Furthermore, in our experiments, we found the performance when using L1 loss and adversarial loss jointly is better than using either of them.
> ------[1] Pathak, D., Krahenbuhl, P., Donahue, J., Darrell, T., & Efros, A. A. (2016). Context encoders: Feature learning by inpainting. In Proceedings of the IEEE Conference on Computer Vision and Pattern Recognition (pp. 2536-2544).
> ------[2] Isola, P., Zhu, J. Y., Zhou, T., & Efros, A. A. (2017). Image-to-image translation with conditional adversarial networks. arXiv preprint.

---

> > ### Comment · AnonReviewer3 · 2018-11-19
> > **good feedback**
> >
> > Dear Authors thank you for your extensive feedback ~
> >
> > I am able to better understand your paper ~ and I believe it would be beneficial to have it at conference.
> >
> > I am thus changing my rating:
> >
> >   Marginally below acceptance threshold  ==> Marginally ABOVE acceptance threshold
> >
> > Thank you!
> >
> > <AnonReviewer3>

---

> > > ### Author Response · Authors · 2018-11-26
> > > **Re: good feedback**
> > >
> > > Thank you!

---

> ### Author Response · Authors · 2018-11-10
> **Clarifications of confused points, and the philosophy behind the architecture: Part II**
>
>
> -----------------------------------------------------------------
> Q: Third, and slightly related to the previous point, why do you need a conditional GAN discriminator, if you already model similarity by L1? Typically one would use a GAN-D() to model "proximity" to the source-distribution, and then a similarity loss (L1 in your case) to model "proximity" to the actual input sample, in the case of traditional domains. Instead, here you seem to suggest using L1 and GAN to do basically the same thing, or with significant overlap anyways. This is confusing to me. Please explain the logic for this architectural choice.
>
> A:(1) The logic of using both of them has been explained in the answer to the last question.
> (2) The logic has been well-utilized and verified in the image-translation domain. Again please see the details in the answer to the last question.
> (3)  Our ablation experiment also demonstrates the similar advantage of using both losses for graph translation than only using L1 loss. Specifically, the proposed GT-GAN that uses both loses outperformed the S-Generator that only uses L1 loss on all three datasets by 10% in accuracy on average as shown in Table 2,3 and 4.
>
> -----------------------------------------------------------------
> Q: Four, could you please explain the setting for the “gold standard” experiment. I'd have to assume, for instance, you train a GNN in a supervised way by using both source (non-suspicious) and target (suspicious) behavior, and label accordingly? That said I am not 100% sure of this problem setting.
>
> A: Yes, “gold standard” method is directly trained based on real target graphs instead of generated ones. Specifically, as you know, all the comparison methods in our paper are generative models which generate graphs, and our experiment is to evaluate how real the generated graphs are. One way to evaluate this is by “indirect evaluation”, where we use the graphs generated by different comparison methods as training data to train a classifier based on KCNN (see reference (Nikolentzos, et al.,2017) in the paper), and then compare which model generates “more-real graphs” by testing their corresponding trained classifier on test set which consists of real graphs. In “gold standard” method, it directly uses the real graphs to train the classifier (still based on KCNN), so it is expected to get the best performance. Therefore, “gold standard” method acts as the “best-possible-performer”, and is used as a benchmark to evaluate all the different generative models on how “real” the graphs they can generate: the closer (and better) their performance is to the “gold standard” one, the “more real” their generated graphs are.
>
> We hope we were able to answer everything to your satisfaction, please let us know if there are any more open points.
>
> Thank you once again!

---

### Official Review · AnonReviewer1 · 2018-11-08
**Novel idea but requesting clarifications.**

**Rating:** 5
**Confidence:** 2

**Review:**

The paper presents a novel idea of generating discrete data such as graphs that is conditional on input data to control the graph structure that is being generated.

Given an input graph, the proposed method infers a target graph by learning their underlying translation mapping by using new graph convolution and deconvolution
layers to learn the global and local translation mapping.

The idea of learning generic shared common and latent implicit patterns across different graph structure is brilliant.

Their method learns a distribution over graphs conditioned on the input graph whilst allowing the network to learn latent and implicit properties.

The authors claim that their method is applicable for large graphs. However, it seems the experiments do not seem to support this.

It is not clear how the noise is introduced in the graphs. I would have expected to see some analysis and results on the translation quality over systematic noise applied to the input graph.

It is also not clear what are the assumptions made on the connectivity of the input graph and the target graph.
Do we know how does the connectedness of the  input graph affect the translation quality in the case of strongly connected directed graphs? Or what happens if the target graph has a strong connectivity? Towards this, how does the computational complexity scale wrt to the connectedness?

A lot of clarity is required on the choice of evaluation metric; for example choice of distance measure ?  What is the L1 norm applied on?

I did not completely follow the arguments towards directed graph deconvolution operators. There is lack of clarity and the explanation seems lacking in parts in this particular section; especially since this is the key contribution of this work

Typo:. The “Inf” in Tabel 1

---

> ### Author Response · Authors · 2018-11-10
> **Clarifications of some points: Part II**
>
>
> -----------------------------------------------------------------------
> Q：A lot of clarity is required on the choice of evaluation metric; for example, choice of distance measure?  What is the L1 norm applied on?
>
> A: Answer about Evaluation metrics:
> (1) We want to evaluate if the generated graphs are scale-free graphs in the direct evaluation for dataset scale-free graphs. If the degree distribution of generated graphs is the same to the degree distribution of real target graphs, the generated graphs are good.
> (2) There are many classical evaluation metrics focusing on measuring the similarities or distance of two distributions. The four metrics in this paper are among the most authoritative and commonly used ones in existing works, e.g., [2][3][4][5].
>
> Answer about L1 norm:
> (1) L1 norm is applied to the weight adjacent matrix of the graph. Our methodology is achieved by a trade-off between L1 loss and adversarial loss (GAN-D). Specifically, L1 makes generated graphs share the same rough outline of sparsity pattern like generated graphs, while under this outline, adversarial loss allows them to vary to some degree.
> (2) L1 norm is commonly used in GAN in relevant domains, e.g., in image-translation domain, for example, reference [1] (with 600+ citations) and reference [6] (with 1300+ citations). They have done extensive experiments to show the advantage of such a strategy. (3) The experiment demonstrates its effectiveness. Specifically, the proposed GT-GAN that uses L1 norm outperformed all the other comparison methods shown in Table 2,3 and 4.
>
> -------[2] Schieber, T. A., Carpi, L., Díaz-Guilera, A., Pardalos, P. M., Masoller, C., & Ravetti, M. G. (2017). Quantification of network structural dissimilarities. Nature Communications, 8, 13928.
> -------[3] Bauckhage, C., Kersting, K., & Hadiji, F. (2015, July). Parameterizing the Distance Distribution of Undirected Networks. In UAI (pp. 121-130).
> -------[4] Chiang, S., Cassese, A., Guindani, M., Vannucci, M., Yeh, H. J., Haneef, Z., & Stern, J. M. (2016). Time-dependence of graph theory metrics in functional connectivity analysis. NeuroImage, 125, 601-615.
> -------[5] You, J., Ying, R., Ren, X., Hamilton, W. L., & Leskovec, J. (2018). GraphRNN: A Deep Generative Model for Graphs. arXiv preprint arXiv:1802.08773.
> -------[6] Isola, P., Zhu, J. Y., Zhou, T., & Efros, A. A. (2017). Image-to-image translation with conditional adversarial networks. arXiv preprint.
>
> -----------------------------------------------------------------------
> Q：I did not completely follow the arguments towards directed graph deconvolution operators. There is lack of clarity and the explanation seems lacking in parts in this particular section; especially since this is the key contribution of this work.
>
> A: (1) Our decoder is symmetric to the encoder in their architectures. The encoder does n-hop edge information aggregation from the input graphs and learns the latent representation of nodes. Then, we first decode the node embedding to get the n-hop aggregated information on edges by node-to-edge layer and then we further decode the n-hop aggregated information layer by layer by n-layers back to get the output adjacency matrix.
> (2) Different from image deconvolution, for each hidden channel, we have two filters vertical to each other, i.e., one is a column vector while the other is a row vector. To get the nth hop information of edge <i,j>, row filter decodes all the (n+1)-th hop information of outgoing edges of node i and column filter decodes all the (n+1)-th hop information of incoming edges of node j.
> (3) To make our description clearer, we have updated our paper in Section 3.2.2, e.g, by adding “To get the nth hop information Aij, row filter decodes all the (n+1)-th hop information of outgoing edges of Vi and column filter decodes all the (n+1)-th hop information of incoming edges of Vj.”
>
> -----------------------------------------------------------------------
> Q: Typo:. The “Inf” in Tabel 1
>
> A: As explained in Section 4.2.4 “Results on Scale-Free Graphs”, the “Inf” in Tabel 1 represents the distance more than 1000.
>
> We really hope that we have explained every confused point clearly and please let us know if there are any other points.
> Thank you once again for your reviews.

---

> ### Author Response · Authors · 2018-11-10
> **Clarifications of some points: Part I**
>
>
> Dear Reviewer:
>
> Thank you very much for your comments and suggestions. We would like to answer your questions in detail as follows:
>
> -----------------------------------------------------------------------
> Q: The authors claim that their method is applicable for large graphs. However, it seems the experiments do not seem to support this.
>
> A: (1) We did not mention that we handle “large graph”, but instead we only mention that we handle “larger” graph. In the domain of graph generation, currently, the proposed graph generative models can typically only deal with graphs with dozens of nodes or less (except GraphRNN which can scale to 300). Compared to them, our model handles relatively “larger graph” (6-10 times larger than most existing methods).
> (2) Translation in graphs is a new topic and we have not found many datasets in very large scale, so we do not test on much larger nodes. But the scalability experiments can still show the superiority of our model compared to others.
> (3) We typically test small-size graphs because most of the comparison methods can only handle small-size graphs.
>
> -----------------------------------------------------------------------
> Q: It is not clear how the noise is introduced in the graphs. I would have expected to see some analysis and results on the translation quality over systematic noise applied to the input graph.
>
> A: Thanks for the review comment.
> (1) The noise is introduced by the dropout function in each convolution layer. Dropout functions by randomly ignore 50% of neuron’s output of a network in our mode by a uniform distribution.
> (2) The way we add noise is well-recognized and commonly-used in generative deep learning models[1]. The noises added in GANs aim to enable the diversities in the generated graphs to avoid the problem that GANs tend to favor producing same output rather than spreading it evenly over the domain.
> (3) We have shown the analysis of the translation quality against noise in Figures 4 and 5. In Figure 5 (see in the supplementary material), each logarithm plot in each column show the power-law trend of each randomly generated graph, which will look linear in such a logarithm plot. It can be seen that the generated graphs show the similar randomness pattern as the real graphs. Moreover, the larger the graph is (see the graph size of 150), the smaller the randomness is, and the clearer the power-law trend is, which verifies that the translation quality of our method.
>   ------[1] Isola, P., Zhu, J. Y., Zhou, T., & Efros, A. A. (2017). Image-to-image translation with conditional adversarial networks. arXiv preprint.
>
> -----------------------------------------------------------------------
> Q: It is also not clear what are the assumptions made on the connectivity of the input graph and the target graph. Do we know how does the connectedness of the input graph affect the translation quality in the case of strongly connected directed graphs? Or what happens if the target graph has a strong connectivity? Towards this, how does the computational complexity scale wrt to the connectedness?
>
> A: (1) Similar to all the existing graph deep generative learning methods for generic graphs, we do not have additional assumptions on the graphs. The domain of graph deep generative learning methods typically do not require to distinguish or preprocess specific topological types of graphs before applying it, no matter it is strongly- or weakly- connected graph, complete graph, planar graph, scale-free graph, or graphs that have other specific patterns. This is actually one of the core advantages of deep learning based models where the graph patterns are not extracted or pre-identified manually by the human but automatically discovered by the end-to-end deep models.
> (2) This paper has given the time complexity in the worst case: O(n^2) as shown in 3.4. The worst case happens when the graph is a complete graph. The time complexity of a strongly-connected graph will not be worse than that.

---

> ### Author Response · Authors · 2018-11-26
> **Recent modifications of Paper1309**
>
> Dear Reviewer,
>
> Thank you very much for your new and previous comments. We have revised our paper again in order to address all of them in the paper. The modifications are listed as followings:
>
> 1. For graph deconvolution, we have modified and reorganized the content. The Section 3.2.2 on “Graph Deconvolution” has been reorganized to two subsections “node-to-edge deconvolution” and “edge-to-edge deconvolution”. We also extended them to make the description on deconvolution operations clearer and more comprehensive.
>
> 2. For graph deconvolution, we have also added a new figure and refined the equations’ descriptions. Figure 3 is added to describe the mechanism of our proposed deconvolution operators as well as their correlation to the convolution operations. Equation 6, Equation 7, and their descriptions have also been revised to make them clearer and concrete. Specifically, Figure 3 describes how the node representation and edge representation are respectively decoded by our deconvolution layers, while Equation 6 and Equation 7 describe how to aggregate the decoded information into the final weighted adjacent matrix.
>
> 3. We have referred to all the figures in the body of text.
>
> 4. We have added statements to describe how to introduce random noises in the whole architecture, see in the 2nd paragraph of Section 3.1 in Page 4.
>
> 5. We have added statements of describing the reason to use L1 loss and how L1 loss is applied, please see in the paragraph before Equation 2 in Page 4. Additionally, we also added the statements of how L1 norm and GAN loss function jointly, see in the paragraph after Equation 2.
>
> 6. We have added the statements why the metrics are chosen to evaluate the scale-free dataset, please see in the 2nd Paragraph of Section 4.2.2.
>
> Additionally, to improve the reproducibility of the proposed methodologies and experiments, we have already released our code in https://github.com/anonymous1025/Deep-Graph-Translation-.  More architecture parameters are also provided in Appendix E.
>
> Thank you very much again for the comments and please let us know if there are any other issues.

---

### Comment · AnonReviewer3 · 2018-11-02
**A few of comments and request for clarifications**

Dear Authors, thank you for your submission.

The problem setting is very interesting, especially the problem of malicious graph activity synthesis "forecast and synthesize the future potential malicious authentication graphs of the users without any historical malicious behaviors, by the graph translator from normal to malicious graph trained based on the users with historical malicious-behavior records.".

That said, I have few points that need clarity:

First, the general architecture, and specifically the logic behind the edge-to-edge convolution, and generally the different blocks in fig.1 "graph translator".

Second, how exactly do you do a L1 loss on graphs? I'd have to assume the topology of the graph is unchanged between Gy and T(Gx) ~ and then maybe take L1 of weight matrix? But then is this general enough ~ given your stated goal of modeling different topologies? Either ways, more explanation / and perhaps equations to clarify this loss would be very helpful.

Third, and slightly related to the previous point, why do you need a conditional GAN discriminator, if you already model similarity by L1? Typically one would use a GAN-D() to model "proximity" to the source-distribution, and then a similarity loss (L1 in your case) to model "proximity" to the actual input sample, in the case of trasductional domains. Instead here you seem to suggest to use L1 and GAN to do basically the same thing, or with significant overlap anyways. This is confusing to me. Please explain the logic for this architectural choice.

Four, could you please explain the setting for the “gold standard” experiment. I'd have to assume, for instance, you train a GNN in a supervised way by using both source (non-suspicious) and target (suspicious) behaviour, and label accordingly? That said I am not 100% sure of this problem setting.

Thank you!

---

### Meta-Review · Area_Chair1 · 2018-12-14
**No reviewer was willing to champion this work**

**Confidence:** 4
**Recommendation:** Reject

**Metareview:**

Although one reviewer recommended accepting this paper, they were not willing to champion it during the discussion phase and did not seem to truly believe it is currently ready for publication. Thus I am recommending rejecting this submission.